# Mutational biases favor complexity increases in protein interaction networks after gene duplication

Angel F Cisneros [ID] [1,2,3,4,5], Lou Nielly-Thibault[2,3,4,6], Saurav Mallik[5], Emmanuel D Levy[5] & Christian R Landry [ID] [1,2,3,4,6] [✉]

## Abstract

**Biological systems can gain complexity over time. While some of these transitions are likely driven by natural selection, the extent to which they occur without providing an adaptive benefit is unknown. At the molecular level, one example is heteromeric complexes replacing homomeric ones following gene duplication. Here, we build a biophysical model and simulate the evolution of homodimers and heterodimers following gene duplication using distributions of mutational effects inferred from available protein structures. We keep the specific activity of each dimer identical, so their concentrations drift neutrally without new functions. We show that for more than 60% of tested dimer structures, the relative concentration of the heteromer increases over time due to mutational biases that favor the heterodimer. However, allowing mutational effects on synthesis rates and differences in the specific activity of homo- and heterodimers can limit or reverse the observed bias toward heterodimers. Our results show that the accumulation of more complex protein quaternary structures is likely under neutral evolution, and that natural selection would be needed to reverse this tendency.**

**Keywords** Gene Duplication; Mutational Biases; Neutral Evolution; Protein Biophysics; Protein Interaction Networks
**Subject Categories** Computational Biology; Evolution & Ecology; Structural Biology

## Introduction

Gene duplication is a potent mechanism for the generation of novel molecular and phenotypic traits (Conant and Wolfe, 2008; Kuzmin et al, 2022). Duplication introduces an additional copy of a gene, which may allow sister copies to accumulate mutations that would not have been accessible before. The long-term maintenance of the duplicates could depend on one or both evolving new functions

(Deng et al, 2010; Bridgham et al, 2008; Kaltenegger et al, 2013; Prakashrao et al, 2022). Alternatively, the two copies could split the functions of the ancestral protein (Force et al, 1999; Lynch and Conery, 2000; He and Zhang, 2005; Baker et al, 2013). Under this model, the two copies show partial degeneration of their ancestral function and complement each other in accomplishing it. In this context, cellular biochemical or genetic networks become more complex (i.e., they have more parts and interactions) without net functional or fitness gain.

One context in which this increase in complexity takes place is the assembly of protein complexes. Following the duplication, two paralogous genes encode the product of the ancestral gene. In the case of the duplication of a gene encoding a homomeric protein complex (self-interacting protein), the paralogous proteins might assemble in a mixture of states (Pereira-Leal et al, 2007; Kaltenegger and Ober, 2015): two homodimers and a heterodimer in the case of dimers, with further combinatorial complexity depending on the number of subunits in the ancestral homomer. In this work, we will focus on homodimers because they are the most frequent type of homomers in proteomes. Indeed, homodimers amount to almost 66% of homomers and no other multimeric assembly exceeds 15% of homomers, with some variation depending on the species (Lynch, 2012, 2013; Levy and Teichmann, 2013; Schweke et al, 2024). An important consideration is that heteromers directly become the state with the highest concentration right after the duplication, and the energetic barrier needed for selective homomerization increases with the number of subunits (Hochberg et al, 2018). For example, the duplication of an ancestral homodimer would lead to 50% heterodimers and 25% of each homodimer at binding equilibrium if all their parameters are the same (synthesis rates, folding energies, binding affinities, etc.). As such, heteromers are expected to assume a significant proportion of the functional contributions directly after the duplication.

Given the prevalence of homomers and the significant role of gene duplications in evolution, it follows that heteromers composed of paralogous proteins constitute a substantial portion of the complexes within protein interaction networks. Indeed, these heteromeric interactions have been shown to be retained in more than 30% of paralogs in several species and are especially prevalent for duplicates with higher sequence identity (Marchant et al, 2019;

[1]Département de biochimie, de microbiologie et de bio-informatique, Faculté des sciences et de génie, Université Laval, G1V 0A6 Québec, Canada. [2]Institut de biologie intégrative et des systèmes, Université Laval, G1V 0A6 Québec, Canada. [3]PROTEO, Le regroupement québécois de recherche sur la fonction, l'ingénierie et les applications des protéines, Université Laval, G1V 0A6 Québec, Canada. [4]Centre de recherche sur les données massives, Université Laval, G1V 0A6 Québec, Canada. [5]Department of Chemical and Structural Biology, Weizmann Institute of Science, 7610001 Rehovot, Israel. [6]Département de biologie, Faculté des sciences et de génie, Université Laval, G1V 0A6 Québec, Canada. ✉E-mail: christian.landry@bio.ulaval.ca

Mallik and Tawfik, 2020). Interestingly, in some of these cases, the heteromer becomes the only active complex (Wang et al, 2006; Boncoeur et al, 2012) or diverges functionally (Bridgham et al, 2008; Baker et al, 2013; Paul and Hristova, 2019; Li et al, 2022). A heteromer taking over the function of the ancestral homomer is an example of degeneration with complementation, whereby a function encoded by a single gene is now encoded by two genes. A consequence of such transitions toward heterodimers is that protein interaction networks do not benefit from the robustness usually brought about by redundancy introduced by duplication (Diss et al, 2017; Dandage and Landry, 2019). Therefore, a major question these observations raise is how and why these transitions do occur.

In particular, here we are interested in understanding whether natural selection is required to drive the evolution from homo-dimers to heterodimers, or whether neutral evolutionary processes alone can account for this transition. Many arguments are in favor of adaptive changes. For example, different homomeric and heteromeric symmetries can facilitate specific properties such as allostery and transport (Bergendahl and Marsh, 2017). There are also compelling and potentially adaptive examples of retained heteromers. For instance, Pillai et al, (2020) thoroughly described the emergence of allosteric cooperativity in heterotetrameric hemoglobin, which is limited in homotetramers of the beta (Kurtz et al, 1981) or gamma subunits (Kidd et al, 2001). Other examples include the BfER steroid receptor that negatively regulates its paralog BfSR (Bridgham et al, 2008), the Mcm1 and Arg80 transcription factors whose homomers and heteromers bind specific targets (Baker et al, 2013), and half-ABC transporters that are only capable of transporting specific molecules out of the cell when forming heterodimers (Boncoeur et al, 2012). In these three examples, heteromerization between paralogs appears to currently be an important element contributing to the new function or feature. However, recent research has shown that protein interac-tion interfaces that do not provide an adaptive benefit can become entrenched as mutations accumulate (Finnigan et al, 2012; Hochberg et al, 2020; Schulz et al, 2022). A notable example is the fungal V-ATPase ring, which after a duplication shifted from a 5:1 stoichiometry to a 4:1:1 stoichiometry, resulting in increased complexity without clear changes in function (Finnigan et al, 2012). Thus, natural selection may not be necessary to explain why many extant heteromers were retained.

The above examples suggest that a fraction of extant heteromers of paralogs provide some adaptive benefit. At the same time, the heteromer of paralogs could have been fixed in a non-adaptive manner before the emergence of the new function. Indeed, different models have been proposed to explain how complexity could arise neutrally (Stoltzfus, 1999; Lynch, 2007; Muñoz-Gómez et al, 2021). In these models, the increase in complexity does not provide a fitness advantage or a disadvantage, but instead tends to become prevalent due to drift and mutational biases. Mutational biases refer to particular types of mutations (or their effects) occurring more often than others, which influence the supply of available mutations for drift and natural selection, and thus ultimately adaptation (Svensson and Berger, 2019; Cano et al, 2022). For instance, in the case of homomers and heteromers, there could be mutational biases that favor the assembly of one type of complex over the other. When everything else is equal, evolution tends to go in the direction of mutational bias.

Deconvoluting the effects of mutational biases in neutral evolution and natural selection requires theoretical null models. Intuitively, if mutations naturally disrupt heteromers or favor homomerization, heteromers would tend to disappear neutrally. On the contrary, if mutations tend to favor the maintenance of the heteromers, then natural selection against these mutations or other mechanisms like compartmentalization and differential regulation would be necessary to separate the proteins in space or time and prevent the assembly of the heteromer (Kaltenegger et al, 2013; Marchant et al, 2019). The effect of individual mutations is expected to be amplified in homomers where they are repeated by symmetry. By contrast, only a subset of the subunits in heteromers would contain a mutation. Hence, since mutational effects on binding are more often destabilizing than stabilizing, one could expect homomers to be more easily disrupted. The inherent difference in expected effects of mutations could be potentiated by any mutational biases in terms of which types of mutations happen more often. Overall, the interplay between the effects of mutations, along with natural selection, should determine the fate of the paralogs. As such, we hypothesize that, in a neutral scenario, heteromers will slowly become more abundant while homomers wither away due to the accumulation of slightly destabilizing mutations.

Here, we use evolutionary simulations to examine the fate of heterodimeric complexes in a context where selection does not differentiate between homo- and heterodimers. Each step of the simulations samples the biophysical effects of mutations to evaluate how they would modify the concentrations at equilibrium of homo- and heterodimers. Natural selection acts on the total amount of protein complexes formed so all three molecular complexes are free to explore a large space of possible binding equilibria neutrally within the limit set by selection. We characterize the relative impact of several parameter values that can modify the proportions of homodimers (HM) and heterodimers (HET), including changes in the synthesis rates and folding energies of individual subunits, as well as the binding affinities and specific activities of their homo- and heterodimers. Finally, we examine the relative advantages that homo- and heterodimers would need to provide to become the dominant type of complex after the duplication.

## Results

### The equilibrium between homomers and heteromers is highly sensitive to mutational effects on binding energy

We built a model that uses several parameters of paralogous proteins (synthesis rates, folding free energies, binding affinities, and decay rates) to estimate the concentrations of monomers and dimers at equilibrium (Fig. 1A). The folding free energies ($\Delta G_{fold}$) of the proteins are used to determine the fraction of folded protein copies (Sailer and Harms, 2017), which can later form homodimers or heterodimers based on their binding affinity ($\Delta G_{bind}$). The concentrations of the three dimers at binding equilibrium are our trait of interest (i.e., [AA], [BB], and [AB]). We can estimate them using the different parameters because at equilibrium the rates of assembly and disassembly of dimers are identical, resulting in a system of equations. The full derivations of the systems of equations used to estimate concentrations at equilibrium before

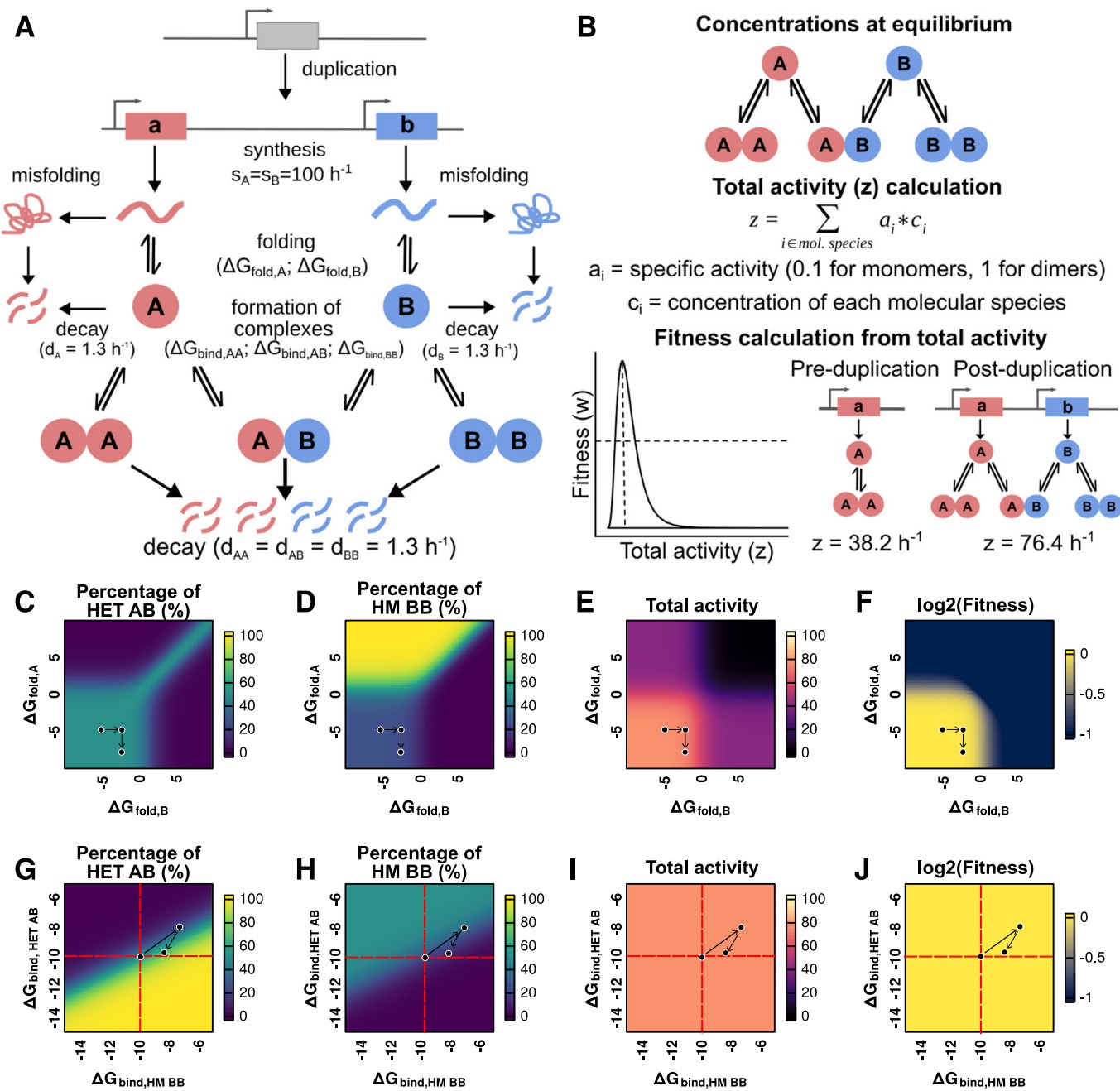

**Figure 1. Changes to folding free energy and binding affinity modify the concentration of each dimer and the fitness of the system.**

(A) Post-duplication system. Paralogous proteins are produced with synthesis rates $s_A$ and $s_B$. Synthesis encompasses both transcription and translation, which are not considered separately. The probability of correct folding is calculated based on the folding free energy ($\Delta G_{fold}$) (see "Methods"). Misfolded proteins are removed from the system. Folded proteins can assemble into homo- or heterodimers based on binding free energies ($\Delta G_{bind,HM\,AA}$, $\Delta G_{bind,HET\,AB}$, $\Delta G_{bind,HM\,BB}$). (B) The concentrations of each molecular species (monomers, homodimers, and heterodimers) are estimated at equilibrium. Total protein activity is calculated by multiplying the equilibrium concentrations by their specific activity (0.1 for monomers, 1 for both homodimers and the heterodimer). Fitness is estimated based on the total protein activity using a lognormal function given by parameters alpha and beta. With the parameters for synthesis rates and decay rates shown in (A), the estimated values of total activity for the system before and after the duplication corresponded to 38.2 h$^{-1}$ and 76.4 h$^{-1}$. (C–F) Effect of variation in the $\Delta G_{fold}$ for each monomer on the percentage of heterodimer (HET AB) (C), percentage of one of the homodimers (HM BB) (D), the total activity (E), and fitness (F) of the system. Fitness is estimated using a lognormal distribution with the optimum (alpha value) set to a total activity of 80 (see (B)). Arrows represent an example of a mutational trajectory. (G–J) Effect of variation in $\Delta G_{bind,HET\,AB}$ and $\Delta G_{bind,HM\,BB}$ on the percentage of the heterodimer (HET AB) (G), percentage of one of the homodimers (HM BB) (H), and the total activity (I), and fitness (J) of the system. (G–J) $\Delta G_{bind,HM\,AA}$ was kept constant at −10 kcal /mol, indicated by the red dashed lines. Arrows represent an example of a mutational trajectory.

and after the duplication are shown in Appendix Notes 1 and 2, respectively. Later, in the evolutionary modeling, these parameters are allowed to evolve through mutations.

To examine how changes in these parameters will determine fitness, we defined a function that links fitness to the concentration of the total amount of protein complex formed (Fig. 1B). Since we are interested in examining how neutral forces contribute to the transition from homomers to heteromers without the need for new functions, selection only acts on the total amount of complex formed (which satisfies the function) and does not distinguish between homo- and heterodimers. These assumptions will be relaxed later, as we include more parameters that could change the contribution of one type of dimer over the other to function.

We characterized the sensitivity of the post-duplication equilibrium concentrations of homo- and heterodimers to changes in the various parameter values. While preserving identical parameter values for both genes after the duplication, the relative concentrations of the three complexes will be 1AA:2AB:1BB. We then examined the post-duplication equilibrium concentration of the three complexes by testing different values for some of the parameters (Fig. 1C–J). Arrows in Fig. 1C–J represent examples of a mutational trajectory to illustrate how changes in the $\Delta G_{fold}$ or $\Delta G_{bind}$ parameters affect the concentrations of complexes at equilibrium, the total activity, and the fitness. We first studied the effects of different combinations of folding free energy values (Fig. 1C–F) around the typical value of $-5$ kcal/mol (Pace, 1975; Plaxco et al, 2000). As expected, destabilizing the fold of only one of the paralogs led to a decrease in the number of available copies of that protein. As a result, the concentrations of its corresponding homodimer and the heterodimer decreased as well, with an increase in the concentration of the other homodimer (Fig. 1C,D). We note that the percentage of heterodimers never exceeded 50% when only folding energies changed, and that it was quickly reduced when either subunit was significantly destabilized. Indeed, destabilizing either or both of the protein subunits ($\Delta G_{fold} >0$) led to a reduction in the total concentration of dimers, and thus in their total activity (Fig. 1E) and fitness (Fig. 1F). Overall, the effect of folding free energy on the balance between homo- and heterodimers was moderate as long as $\Delta G_{fold} < 0$ for both subunits.

We then studied the effects of changes to binding affinity in their typical range of values, centered around $-10$ kcal/mol (Choi et al, 2015; Jankauskaitė et al, 2018). Figures 1G–J show how changes in the binding affinities of the heterodimer ($\Delta G_{bind,HET\ AB}$) and one of the homodimers ($\Delta G_{bind,HM\ BB}$) impact the overall system. Throughout this analysis, the binding affinity of the second homodimer ($\Delta G_{bind,HM\ AA}$) is kept constant at $-10$ kcal/mol to allow representing the data in two dimensions. As expected, the heterodimer was enriched if it became the complex with the strongest binding affinity and depleted if it had the weakest binding affinity (Fig. 1G,H; Appendix Fig. S1A–D). Interestingly, there are regions in the solution space in which most of the available subunits assembled into homodimers although one of the homodimers had a slightly weaker binding affinity than the heterodimer (Appendix Fig. S1E). In this case, equilibrium favors the strongest homodimer. As a result, heterodimers are depleted in the system and the second protein forms its respective homodimer. On the contrary, heterodimers were enriched when their binding affinity was slightly weaker than that of one of the homodimers but

much stronger than that of the second homodimer (Appendix Fig. S1F). Here, any exchange of subunits between the heterodimer and the stronger homodimer would release subunits of the weaker homodimer, leading to an imbalance of concentrations of free subunits. Thus, equilibrium would favor strong heterodimers over a mixture of strong and weak homodimers. Importantly, in the range of values tested and assuming the specific activities of homo- and heterodimers are identical, the total activity and fitness of the system remain unchanged despite shifts in the concentration of each dimer (Fig. 1I,J). The effects of an extended range of values of binding energy are shown in Appendix Fig. S2. When one of the homodimers is completely destabilized, the system is dominated by the equilibrium between the second homodimer and the hetero-dimer (Appendix Fig. S2A,B). In this case, if the heterodimer is also destabilized, the remaining homodimer can approach 100% of the dimers. Such destabilization would lead to a return of the pre-duplication state of only one homodimer, with a corresponding loss in total activity (Appendix Fig. S2C) and fitness (Appendix Fig. S2D) of the system. Overall, the post-duplication equilibrium is highly sensitive to changes in binding affinity when all other parameters remain constant: there is a narrow region where the concentrations of homo- and heterodimers stay close to the original 1AA:2AB:1BB ratios, but small changes can lead to drastic changes in the concentration of each dimer.

The solution space of the system of equations shows that directly after duplication, the equilibrium concentrations of each dimer are particularly sensitive to mutational effects on binding affinity. Differences between the binding affinity of homo- and hetero-dimers in the range of 0.5–1.0 kcal/mol are sufficient to drastically alter the representation of the heteromer in the system (Fig. 1G). This is supported by empirical observations. Studies show that homomeric specificity can be achieved with just 1–3 substitutions (Ashenberg et al, 2011; Garcia-Seisdedos et al, 2017; Stutz and Blein, 2020; Emlaw et al, 2021) and marginal differences in binding affinity (Hochberg et al, 2018). On the other hand, there are regions of the solution space that are much less sensitive to these mutational effects, for example, when the difference in $\Delta G_{bind}$ between the homodimers and the heterodimer is already very large. These results are in agreement with previous models on the energetic cost of homomerization derived using a different approach (Hochberg et al, 2018). An important consideration is that these changes in binding affinity alter the representation of each dimer in the system, but do not modify the total concentration of complexes, as seen in Fig. 1I. Thus, there would be many ways for the cell to provide the same total concentration of complexes, all of which could have the same fitness if every dimer has the same specific activity (Fig. 1J). While the landscapes presented on Fig. 1 help visualize the effect of changes to one parameter at a time, the fate of modern paralogs (Marchant et al, 2019; Mallik and Tawfik, 2020) is a product of the co-evolution and correlation of many parameters (synthesis, folding, binding, activity, etc.). Simulating evolution of these landscapes therefore needs to be based on the actual distributions of effects, which can vary from one protein to another. Importantly, these observations show that the system could theoretically deviate strongly from the starting relative concentrations while keeping the same overall performance and this, in a directional manner, if mutations have biased effects in one direction or the other.

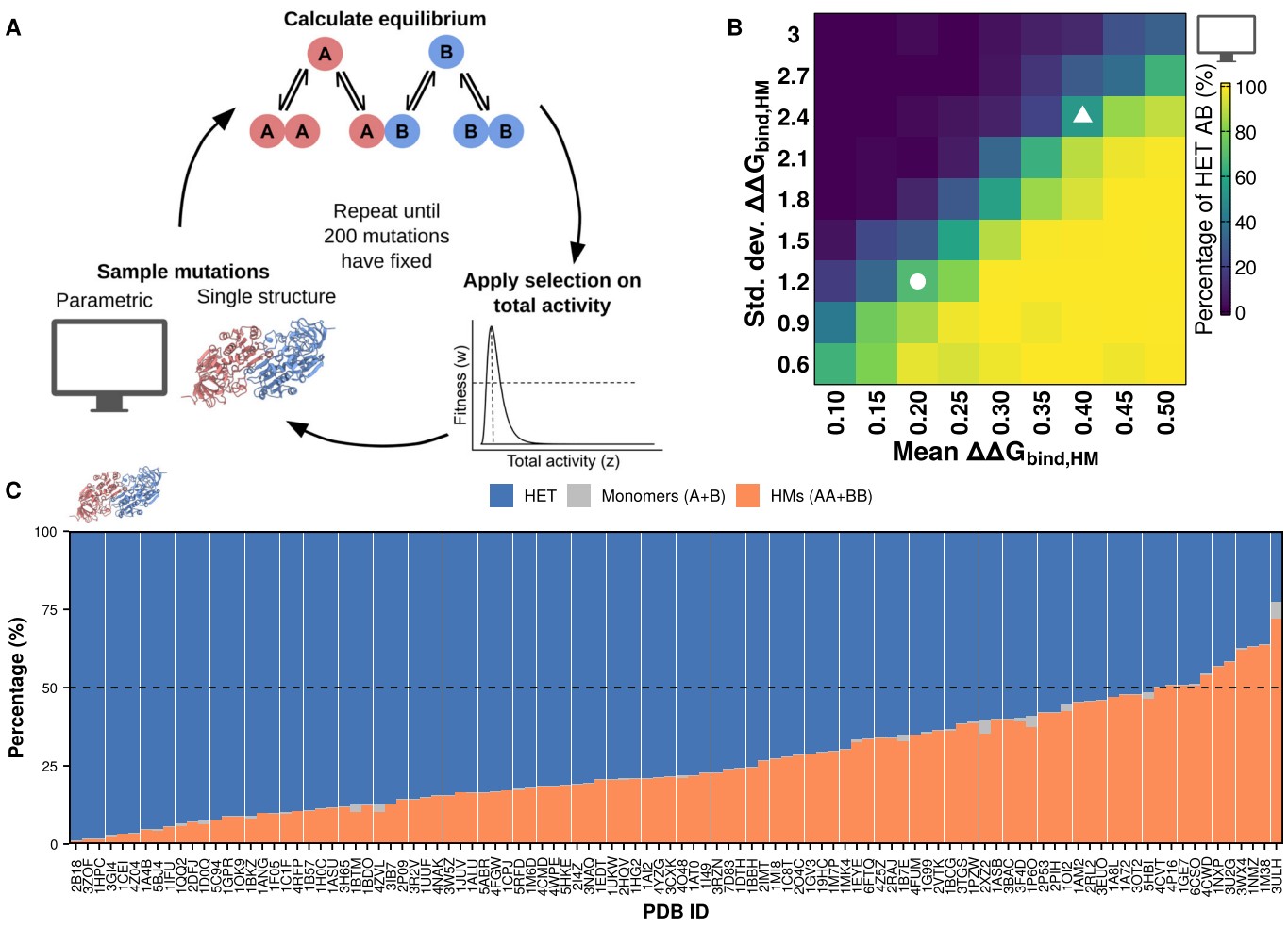

**Figure 2. Biases in the distribution of effects on mutations on binding affinity can drive enrichments of homomers or heteromers.**

(**A**) Overview of the simulation workflow. Homodimers from the PDB were used to infer distributions of mutational effects for two regimes of simulations: those with distributions based on the pooled data (parametric) and those using data for a single structure. Mutations are sampled from these distributions, after which the concentrations of dimers and monomers at equilibrium are recalculated. Selection is applied based on the total activity in the system. Simulations continued until 200 mutations were fixed. The structure for 1A72, used as the cartoon for the single structure simulations, was visualized with ChimeraX (Pettersen et al, 2021). (**B**) Percentage of heteromers at the end of the simulations with each set of parameters for the homodimers (average of 50 replicates, 200 mutations fixed for each). The white circle indicates the condition in which $\Delta\Delta G_{bind,HET}$ and $\Delta\Delta G_{bind,HM}$ are equally distributed ($N$ (0.2, 1.2)). The white triangle indicates the condition in which the distribution for $\Delta\Delta G_{bind,HM}$ uses the same parameters calculated when considering all the sampled structures ($N$(0.4, 2.4)). (**C**) Average percentages of heterodimers, homodimers, and monomers for each of the PDB structures at the end of the simulations. The dashed line indicates the starting point at 50% heterodimers and 50% homodimers (25% of each homodimer).

## Evolutionary simulations show a general bias toward heterodimers replacing the ancestral homodimer

We used the above model to simulate the evolution of the post-duplication equilibrium between homo- and heterodimers as mutations accumulate (Fig. 2A). Mutations lead to changes in the protein sequence, which can affect the binding energy and the folding free energy of proteins. To derive the distributions of mutational effects, we sampled a set of 104 homodimers with crystal structures from the Protein Data Bank (Berman et al, 2000) and simulated the effects of all possible single amino acid substitutions on folding free energy and binding affinity with FoldX (Delgado et al, 2019) (see "Methods"). Briefly, entries from the PDB were chosen if their structures were solved by X-ray

crystallography, annotated as homomers, having identical subunits, and with sequence lengths between 60 and 450 residues. From the resulting set of structures, we selected entries from the major ECOD architecture types (Cheng et al, 2014) to include at least seven for each of them. Similarly, we clustered these proteins at 40% sequence identity with CD-Hit (Li and Godzik, 2006; Fu et al, 2012), and selected one representative protein from each cluster.

We used our model and the dataset of structures to simulate the post-duplication evolution of pairs of paralogs. We considered two different regimes for our simulations: parametric, in which we sampled mutational effects from multivariate normal distributions with different parameters based on the pooled structural data; and single structure, in which mutational effects are sampled directly from the distribution of effects of the corresponding structure. Since mutations

can affect both binding affinity and folding free energy, we derive multivariate normal distributions for the effects of mutations based on the pooled data. These distributions have the following parameters, all in kcal/mol: for $\Delta\Delta G_{bind, HET}$, mean = 0.2, standard deviation = 1.2 (denoted as $N(0.2, 1.2)$); for $\Delta\Delta G_{bind, HM}$, $N(0.4, 2.4)$; and for $\Delta\Delta G_{fold}$, $N(2.6, 4.6)$. These FoldX-derived data also revealed the extent of correlations between mutational effects on folding energy and binding affinity: $r = 0.9$ between $\Delta\Delta G_{bind, HET}$ and $\Delta\Delta G_{bind, HM}$ (Fig. EV1A), $r = 0.3$ between $\Delta\Delta G_{bind, HM}$ and $\Delta\Delta G_{fold}$ (Fig. EV1B), and $r = 0.3$ between $\Delta\Delta G_{bind, HET}$ and $\Delta\Delta G_{fold}$ (Fig. EV1C). As such, we imposed these correlations in our parametric simulations. We annotated figure panels for results for parametric simulations with a computer cartoon and results for single structure simulations with a protein dimer cartoon throughout the rest of the text.

Each step of the simulation represents a mutational event. Importantly, the underlying distribution from which the effects of mutations are sampled is kept constant. As discussed later, we apply this simplifying assumption to keep the simulations scalable because, while the effects of new individual mutations are contingent on previous ones, it is unclear whether the shape of the global distribution changes. Once a mutational effect is sampled, we update all relevant binding affinities and folding energies and recalculate the concentrations at equilibrium for monomers and dimers. Next, we use the equilibrium concentrations to obtain the total activity of the system. Fitness is then estimated using a lognormal function given by two parameters: alpha (the value of total activity that maximizes fitness) and beta (the fitness corresponding to activity values of half and twice the value of alpha). We tested our simulations with two different values for alpha to consider two different scenarios: one in which the total activity after the duplication ($76.4 \, h^{-1}$) is very close to the optimal value (alpha = $80 \, h^{-1}$), and one in which it overshoots the optimal value (alpha = $60 \, h^{-1}$). Beta was set to 0.5 for all simulations. Finally, fitness is used to calculate the probability of fixation for that mutational effect. Once a mutational effect fixes or is rejected, the simulation continues with the next sampling step. Each simulation was run with 50 replicates and allowed to continue until 200 mutations were fixed. Some of the tested proteins are shorter than 200 residues (mean = 240, interquartile range = [157.8, 332], minimum length of 66). Although this implies that 200 mutations could be enough to mutate every position in the sequence, the sequences of paralogs maintained about 50% identity with the WT sequence and more than 30% sequence identity with one another by the end of the simulations. These measures are consistent with real-world scenarios where more than half of *Saccharomyces cerevisiae* and *Escherichia coli* paralogous pairs exhibit ≥30% identity (Mallik and Tawfik, 2020) and suggest our model captures structural constraints in the relative mutability of different positions. Since the results for the two scenarios (alpha = $60 \, h^{-1}$, alpha = $80 \, h^{-1}$) generally agreed, the results with alpha = $80 \, h^{-1}$ are shown in the main text and those with alpha = $60 \, h^{-1}$ are shown in the expanded view.

We tested how changes in the parameters of the distribution of mutational effects of the homodimer affected the simulation outcome (Fig. 2B; Appendix Fig. S3). We allowed the parameters for the distribution of effects on $\Delta\Delta G_{bind, HM}$ to change while the distribution for $\Delta\Delta G_{bind, HET}$ was given by $N(0.2, 1.2)$ and the distribution for $\Delta\Delta G_{fold}$ was given by $N(2.6, 4.6)$. As expected, homodimers dominate when the average mutation is more

destabilizing for the heterodimer than for the homodimer. Homodimers also dominate when the variance in mutational effects on $\Delta\Delta G_{bind, HM}$ is high, since this allows for extreme stabilizing effects on the homodimer. High variance also allows extreme destabilizing effects to be sampled often but they would be likely eliminated by selection because they also affect the total concentration of the complexes. Interestingly, simulations with parameters along the diagonal of Fig. 2B tend to maintain both homo- and heterodimers, with two conditions of particular interest indicated with a circle and a triangle. The circle indicates the simulation in which mutational effects are the same for homo- and heterodimers, while the triangle represents the simulation in which mutational effects on the homodimer are double of those on the heterodimer as in our data derived with FoldX. The preservation of the mixture of homo- and heterodimers in these two conditions suggests that the mere doubling of mutational effects on the homodimer does not lead to an increase in the concentration of heterodimers, as long as effects are sampled from correlated normal distributions. Thus, our parametric simulations show that when none of the dimers offers an advantage over the others, biases in the distribution of available mutational effects alone can determine which dimer dominates at the end of the simulation.

We next set out to test our simulation system with the direct distributions of mutational effects for each of the PDB structures sampled. Simulations were run with the estimated distributions of mutational effects for 104 structures of homodimeric proteins. Replicates of the same simulation followed different trajectories due to the inherent stochasticity of the process. Nevertheless, we observed that different protein structures were associated with different average outcomes, implying that structural features can encode a homodimer- or heterodimer-dominant fate after duplication (Fig. 2C; Appendix Fig. S4). Strikingly, for most of the protein structures tested, the average concentration of heterodimers at the end of the simulations was higher than the starting 50%. Indeed, for 66 structures (63.4%) the final concentration of heterodimers was higher than 70%, as opposed to only 1 structure (1%) for which the final concentration of homodimers was higher than 70%. For the remaining 37 structures (35.5%), the final concentrations stayed closer to the initial equilibrium, with neither heterodimers nor homodimers enriched beyond 70%. The enrichment of heteromers tended to occur relatively early in the simulations, with some structures showing high heterodimer concentrations after fewer than 100 mutations had been fixed (Appendix Fig. S4). This is consistent with our previous finding that heteromerization also dominates in recently duplicated yeast paralogous pairs that have retained high sequence identity (Marchant et al, 2019; Mallik and Tawfik, 2020). These observations suggest there exists a neutral bias toward the replacement of ancestral homodimers by heterodimers, although independent identical events might reach different outcomes.

In our previous simulations, the optimum value for total activity (alpha = $80 \, h^{-1}$) was slightly higher than the total activity directly after the duplication. The results of simulations where the duplication overshoots the optimum (alpha = $60 \, h^{-1}$) correlated well ($r = 0.85$), although simulations with alpha = $60 \, h^{-1}$ consistently yielded slightly lower percentages of heterodimers (Fig. EV2A,B). Considering that the post-duplication total activity is $76.4 \, h^{-1}$, simulations with alpha = $60 \, h^{-1}$ required a reduction of total activity to approach the optimum fitness. Indeed, at least one

of the subunits was quickly destabilized in these simulations (Fig. EV2C), which did not happen in the simulations with alpha = 80 h$^{-1}$ (Fig. EV2D). The quick destabilization observed in the simulations with alpha = 60 h$^{-1}$ occurred at the same time as an initial decrease in the concentration of heterodimers since the available concentrations of free subunits would be different (Fig. EV2E). As the simulations advanced, the concentrations of heterodimers increased, both when alpha = 60 h$^{-1}$ and when alpha = 80 h$^{-1}$ (Fig. EV2E,F). Thus, our simulations show the inherent bias toward the increase in the concentration of heterodimers is robust to the position of the fitness optimum.

These results motivated us to identify structural properties that may influence the outcome of the simulations. First, we focused on the interfaces of our set of PDB structures. We looked at the relative size of the interface core (Appendix Fig. S5A), the overall stickiness of the interface (Appendix Fig. S5B), the proportion of interface core residues that are involved in homotypic contacts (Appendix Fig. S5C), and the overall secondary structure of the interface (Appendix Fig. S5D). However, none of these interface parameters showed a clear association with the outcome of the simulations. We extended our secondary structure analysis to the ECOD architectures (Cheng et al, 2014) of our protein set (Appendix Fig. S5E; Dataset EV1). Outcomes of simulations with structures with similar ECOD architectures were highly variable, suggesting that the general structural parameters we tested are not sufficient to make an association with the bias toward heterodimers. Similarly, we observed that modifying the starting parameters of subunit folding free energy and binding affinity tended to have little effect on the most frequent outcome of simulations with a given structure (Appendix Fig. S6). These results suggest that the outcome of the simulations depends on properties of the distribution of mutational effects that are not directly derived from these structural features.

## Weak but constant mutational biases influence the outcome of evolution

We reasoned that, although the distributions for all proteins in our set are very similar, they might differ with respect to the availability of outliers or pervasive biases with small magnitude in terms of folding or binding energies, which would then drive evolution in one particular direction. Considering that changes in $\Delta G_{fold}$ never lead to an increase in the concentration of heterodimers (Fig. 1C), we focused our analysis on the effects on $\Delta G_{bind}$. For simplicity, we first compared the distributions of available and fixed mutational effects on $\Delta G_{bind,HET}$ and $\Delta G_{bind,HM}$ for the structures with the two most extreme outcomes (2B18 as heterodimer-dominant and 3ULH as homodimer-dominant) (Fig. 3A,B). As noted above, mutational effects on $\Delta G_{bind,HET}$ and $\Delta G_{bind,HM}$ are highly correlated for both structures, and fixed mutations were not restricted to any single part of the distribution. Thus, we hypothesized that proteins that are enriched for heterodimers in the simulations might have access to more mutations with slight deviations from the expected diagonal that favor the heterodimer, which would gradually accumulate as the simulation progresses. For example, a mutation that is below the diagonal in Fig. 3A,B would destabilize the heterodimer less (or stabilize it more) than expected based on the effect on the homodimer, whereas mutations above the diagonal would favor the homodimer. We calculated these deviations from

the diagonal as follows:

$$residual = \Delta\Delta G_{bind,HETAB} - 0.5 * \Delta\Delta G_{bind,HMAA}$$

The cumulative distributions of residuals of available and fixed mutations show minor differences overall (Fig. 3C,D). While comparing 2B18 (the most heterodimer-dominant structure) to 3ULH (the most homodimer-dominant structure) reveals that 3ULH has a slightly higher density of mutations with positive (homodimer-favoring) residuals, the effect is not as clear for the rest of the structures. To have a finer resolution, we defined an enrichment of heterodimer-favoring residuals as the percentage of residuals with magnitudes below −0.2 kcal/mol (heterodimer-favoring) minus the percentage of residuals with magnitudes above 0.2 kcal/mol (homodimer-favoring). Structures with a higher enrichment of heterodimer-favoring mutations are the ones with the highest concentrations of heterodimers at the end of the simulations (Fig. 3E,F). We note that the available enrichment of heterodimer-favoring mutations is amplified in the mutations that fix throughout the simulations. The results of repeating this analysis with other thresholds for the residuals are shown in Table EV1 ($0.39 < r < 0.54$ for available mutations, $0.67 < r < 0.83$ for fixed mutations). Indeed, the cumulative sum of residuals fixed up to a particular point in the simulations explains very well the observed percentage of heteromers (Appendix Fig. S7). We repeated the previous analyses with only the structures evaluated as very high quality in the QSBio database (37 out of 104 structures) (Dey et al, 2018). Simulations with these structures also led to an enrichment of heterodimers due to mutational biases (Fig. EV3), although the increase in heterodimer concentration was less drastic than in the full dataset. As a result, small but consistent mutational biases appear to be responsible for the increase in the concentration of the heterodimer, even without selection explicitly favoring it.

## Differences in synthesis rate and specific activity can reverse the bias toward heterodimers

Our results so far show that the concentration of heterodimers relative to the homodimers after duplication is expected to increase even in the absence of selection favoring this complex over the homodimers. However, in nature, a large fraction of duplicated homodimeric proteins do not maintain their heterodimers. Other forces could thus counterbalance this inherent mutational trend. These include differences in compartmentalization, expression profiles, and specific activities. Paralogs that are separated into different compartments, such as yeast thioredoxin reductases TRR1 (cytosolic) and TRR2 (mitochondrial), do not interact in vivo despite still being capable of doing so in vitro (Oughtred et al, 2021; Mallik et al, 2022). For differentially expressed paralogs, such as E. coli LYSS (constitutive) and LYSU (heat-induced) (Brevet et al, 1995), equilibrium shifts toward the homodimer of the more highly abundant protein. Differences in the specific activity of homomers and heteromers would help selection distinguish between the two of them to maintain a particular total activity in the system.

We therefore examined the impact of differential synthesis rates and specific activities on the outcome of evolution. We do not model the case where one of the dimers could evolve a novel function, because it

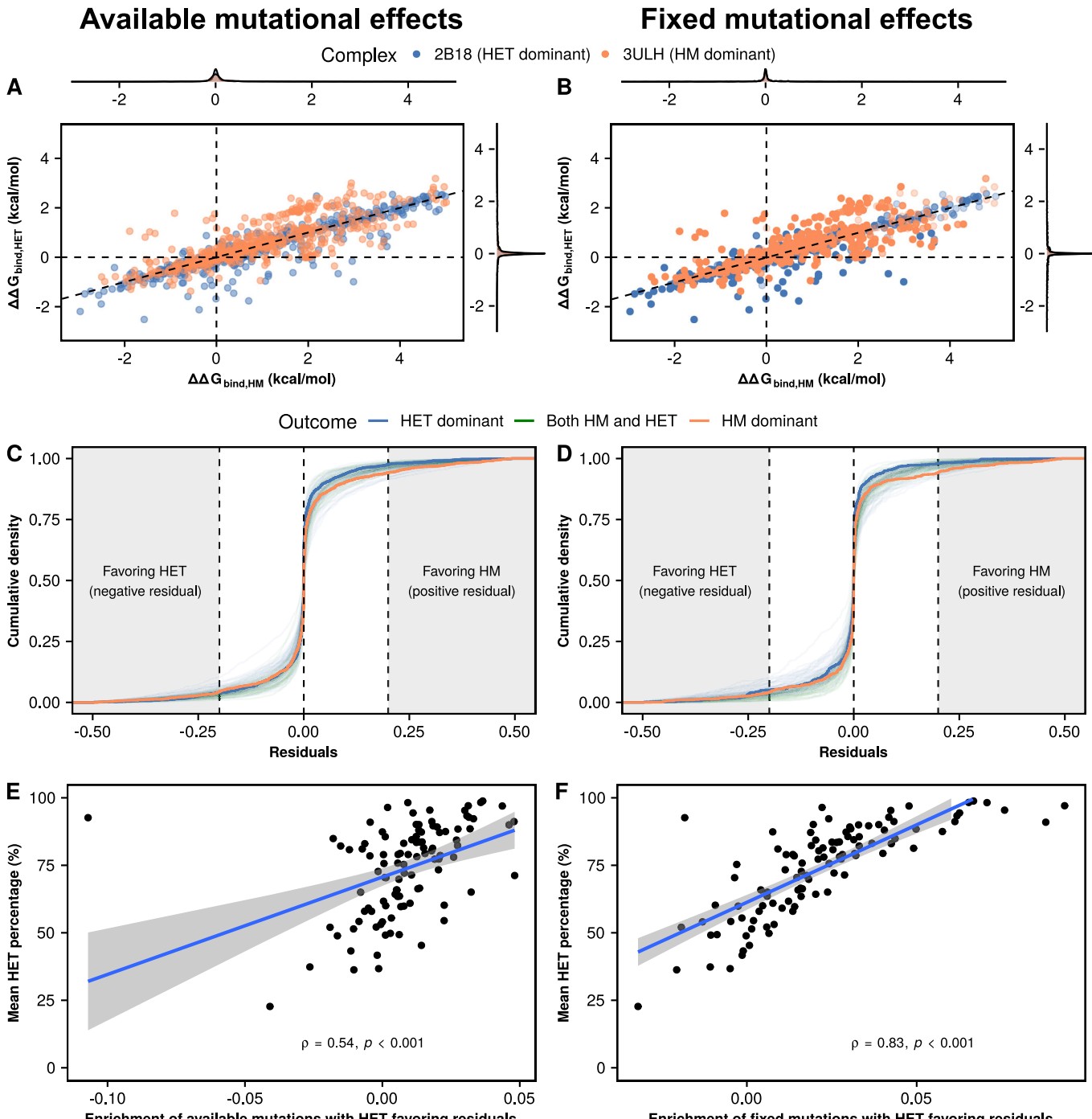

Figure 3. Slight deviations from the diagonal (residuals) are associated with the outcome of the simulations.

(A, B) Distributions of available (A) and fixed (B) mutational effects on $\Delta G_{bind,HET}$ and $\Delta G_{bind,HM}$ for the two proteins with the most extreme outcomes in Fig. 2C: 2B18 (heterodimer-dominant) and 3ULH (homodimer-dominant). The diagonal ($\Delta\Delta G_{bind,HET} = 0.5 * \Delta\Delta G_{bind,HM}$) is shown to illustrate the expectation of mutations on the heterodimer having half of the effect of mutations on the homodimer. (C, D) Distributions of available (C) and fixed (D) residuals with respect to the expectation for each of the 104 structures. Colors indicate mean outcomes for each structure: HET dominant (heterodimer >70% of dimers), HM dominant (homodimer >70% of dimers), both HM and HET (rest of cases). The thicker blue and orange lines represent 2B18 and 3ULH, respectively. Shaded regions on the left and right indicate mutations whose residuals have a magnitude of at least 0.2 kcal/mol favoring either the heterodimer (negative values) or the homodimer (positive values). (E, F) Correlation between enrichment of available (E) and fixed (F) mutations with residuals favoring the heterodimer and the mean percentage of heterodimers at the end of the simulations. The enrichment of heterodimer-favoring residuals is calculated as the density of mutations with residuals smaller than −0.2 kcal/mol minus the density of mutations with residuals greater than 0.2 kcal/mol. P values for (E, F) are calculated using the asymptotic t approximation method for Spearman correlation coefficients.

would require multiple additional considerations. For example, the system would need to describe and deconvolute mutational effects on each of the two functions. Also, the two functions could have different weights in the fitness calculations. Additionally, those considerations should be specific to each protein, unlike folding and binding which are common physical properties of all the dimers we consider.

We first used our equations to characterize the effect of changes in relative synthesis rates between paralogs. We observed that the concentration of the heterodimer is maximized when synthesis rates were identical (Fig. 4A). Conversely, the concentration of either homodimer was maximized as the synthesis rate of its subunits increased beyond that of its paralog (Fig. 4B). The total activity of the system increased with the synthesis rate of one of the paralogs (Fig. 4C), which initially led to higher fitness as total activity approached the optimum of our fitness function (alpha = 80 h$^{-1}$). However, fitness decreased as total activity went beyond the optimum (Fig. 4D). Since many paralogous proteins are differentially expressed (Gout et al, 2010; Gout and Lynch, 2015; Aubé et al, 2023), expression divergence could counteract mutational biases that favor heterodimers. However, the extent of this effect would depend on the relative frequencies of mutational effects on synthesis rates versus the coding sequence, as well as the fitness effects of changes in the total activity of the system due to the total amount of protein copies produced.

Mutations can affect synthesis rates in multiple ways. For example, they could affect binding of transcription factors to a promoter or translation rates (Hausser et al, 2019; Verma et al, 2019; Kemble et al, 2020; Roos and de Boer, 2021; Aubé et al, 2023). Here, we did not distinguish between each of these individual mechanisms but considered synthesis as encompassing all possible effects. We considered the distribution of the effects of mutations on gene expression observed by Metzger et al, (2016) for the TDH3 promoter. We added a new parameter representing the probability of mutations affecting synthesis rates ($p_{exp}$) in the simulations. In these cases, we sampled the effect of mutations on synthesis rates from a separate skew-normal distribution (mean = 0, variance = 0.025, skew = −0.125) following the Metzger et al, (2016) data. These effects were considered to be multiplicative, i.e., the current synthesis rate at the corresponding step in the simulations would be multiplied by the value sampled from the distribution of mutational effects on synthesis rates.

Since promoter sequences can have widely variable rates of evolution (Young et al, 2015, 2022), we proceeded to repeat our simulations with different probabilities of mutational effects affecting synthesis rates. In these parametric simulations, we observed that higher probabilities of mutations affecting synthesis rates generally led to less extreme outcomes in terms of homodimer or heterodimer dominance (Fig. 4E; Appendix Fig. S8), even when mutational effects on binding affinity were skewed against either of them. At higher probabilities of mutations affecting the synthesis rate, the concentration of the heterodimer remained in the range between 40 and 60% of the complexes. Similarly, the bias toward heterodimers observed for the simulations with individual PDB structures became considerably weaker as the probability of mutations affecting synthesis rates increased (Fig. 4F). We note that the synthesis rates of the two paralogs did not diverge considerably in the simulations (in the most extreme case, with mutations having $p_{exp} = 0.9$, median divergence in synthesis rates for the same replicate = 26.1 h$^{-1}$, interquartile range = [12.07 h$^{-1}$,

44.6 h$^{-1}$]) (Appendix Fig. S9). Such differences in protein abundance represent a log2 fold change of 0.35, which is within the range observed for paralogous proteins (Aubé et al, 2023). We repeated these simulations setting the optimal activity value (alpha) to 60 h$^{-1}$ so that the duplication would overshoot the optimum and obtained similar results, albeit with slightly lower concentrations of heterodimers (Fig. EV4).

Overall, allowing changes in synthesis rates tended to preserve both homo- and heterodimers. This effect can be understood as two components. First, divergence in synthesis rates would naturally shift equilibrium away from the heterodimer and toward the homodimer of the more abundant protein (Fig. 4A,B). Since fitness was sensitive to synthesis rates (Fig. 4D), it was likely that an increase in the synthesis rate of one protein would be followed by a decrease in the synthesis rate of its paralog, even if these differences in synthesis rates were moderate (Appendix Fig. S9). Second, because mutations in our simulations either affect synthesis rates or the protein sequence, accelerating the evolution of synthesis rates implied decelerating the evolution of the coding sequence and limiting the accumulation of mutational biases that led to the enrichment of the heterodimer. Our result that allowing changes in synthesis rates limits the bias toward heterodimers can help explain why extant paralog pairs often keep both homomers and heteromers (Marchant et al, 2019; Mallik and Tawfik, 2020), considering that the abundances of paralogs often diverge (Gout et al, 2010; Gout and Lynch, 2015; Aubé et al, 2023).

We then characterized the effect of changes in the specific activity of the homodimers or the heterodimers. In real biological systems, the expectation would be that these specific activities start out as being identical and then slowly diverge as mutations accumulate. However, since the relationship between sequence and function is complex and specific to each protein, we decided to use the simplifying assumption that specific activities would differ directly from the beginning. In these simulations, the more active dimer always has a specific activity of 1, whereas the less active dimer has a specific activity between 0.2 and 0.9. We define a HET activity bias as the difference in specific activities. For example, a −10% HET activity bias indicates the heterodimer is less active than the homodimers (specific activities of 0.9 for heterodimers and 1 for homodimers), whereas a 10% HET activity bias indicates the heterodimer is more active (specific activities of 1 for heterodimers and 0.9 for homodimers). As expected, changes in the specific activity did not affect the assembly of complexes (Fig. 5A,B), since it is determined by binding affinities and the concentrations of properly folded subunits. However, they did have an impact on the total activity of the system (Fig. 5C) and fitness (Fig. 5D). Total activity was maximized when the most active dimer was the most abundant one (top left and bottom right corners of Fig. 5C). Since the specific activity of homodimers or heterodimers never exceeded 1 with our definition of HET activity bias, fitness was maximized in the same regions where activity was maximized (Fig. 5D), although the landscape would be different if we allowed values higher than 1. Thus, the solution space of the system of equations shows that allowing differences in specific activity can promote selection for either homodimers or heterodimers.

We then tested the effect of allowing different specific activities for homo- and heterodimers in our simulations of evolution. Our parametric simulations show that differences in specific activity can overturn the bias toward heterodimers when the optimum is set to

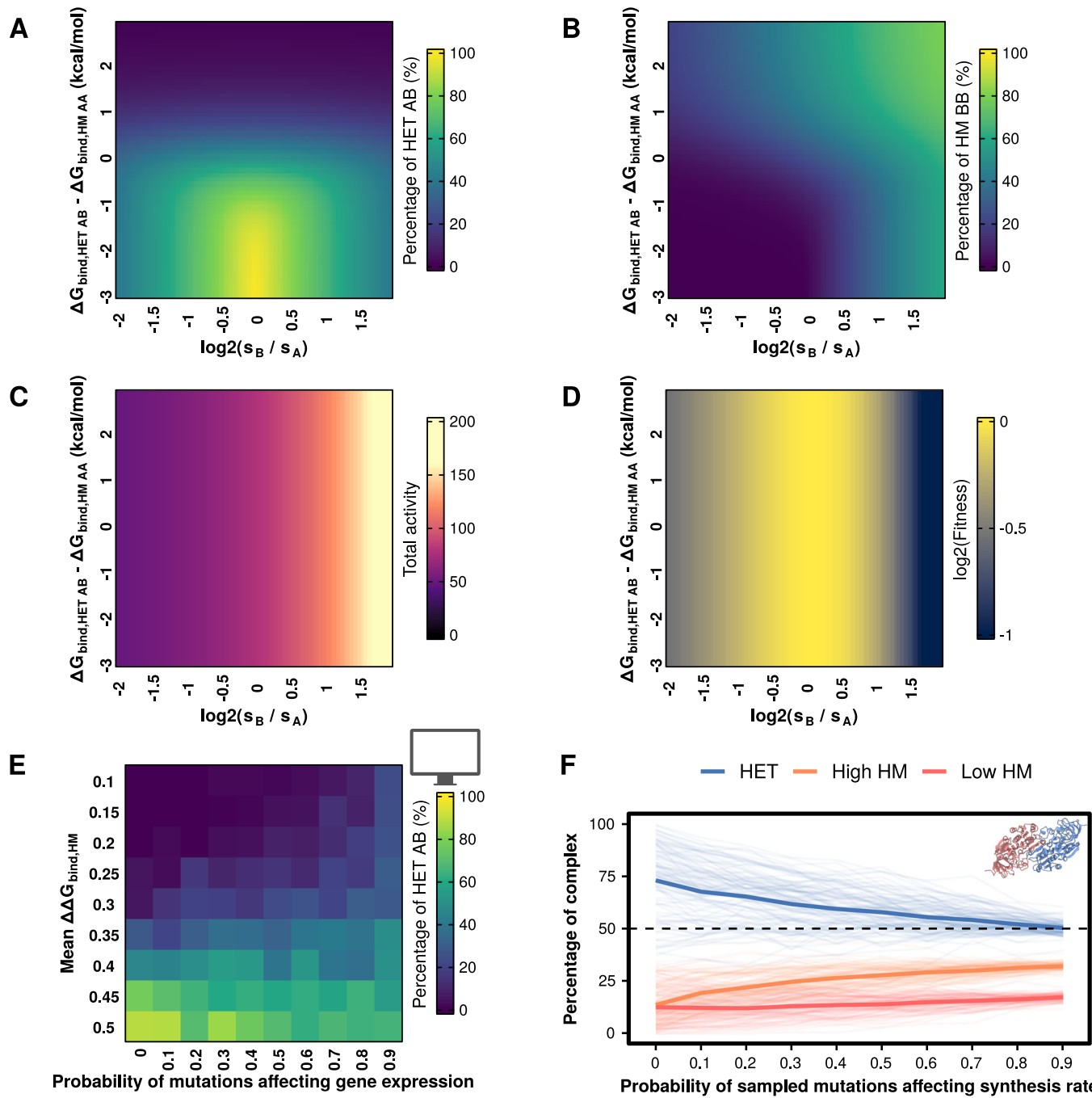

**Figure 4. Allowing changes in synthesis rates can limit the bias toward heterodimers.**

(A–D) Percentage of heterodimers (A), one of the homodimers (B), total activity (C), and fitness (D) as a function of the ratio of synthesis rates and differences between $\Delta\Delta G_{bind,AB}$ and $\Delta\Delta G_{bind,HM}$. (A–D) $s_A$, $\Delta\Delta G_{bind,AA}$, and $\Delta\Delta G_{bind,BB}$ remain constant throughout. Changes in the y-axis indicate changes in $\Delta\Delta G_{bind,AB}$, with all three values ($\Delta\Delta G_{bind,AA}$, $\Delta\Delta G_{bind,AB}$, $\Delta\Delta G_{bind,BB}$) being identical at zero. (E) Percentages of heterodimers at the end of parametric simulations allowing changes in synthesis rates and varying mean effects of mutations on homodimers. (F) Percentages of heterodimers and each of the two homodimers for each of the 104 tested structures when simulations include different probabilities of mutations affecting synthesis rates. The two homodimers were distinguished based on their abundance, such that high HM refers to the one with the higher concentration and low HM to the one with a lower concentration.

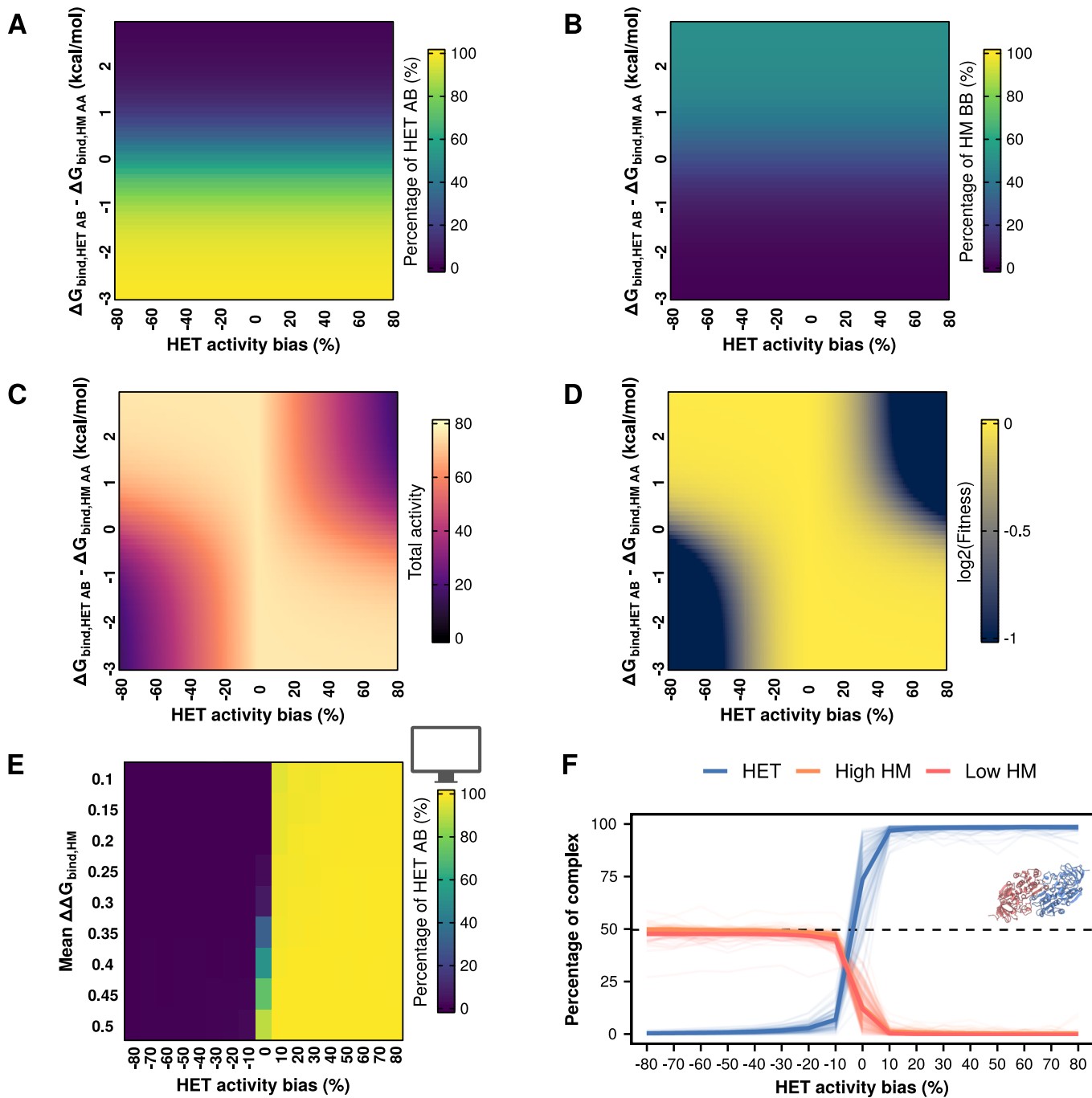

**Figure 5. Differences in specific activities of homodimers and heterodimers can revert the observed bias towards heterodimers.**

(A–D) Percentage of heterodimers (A), one of the homodimers (B), total activity (C), and fitness (D) as a function of differences in specific activity and differences between between $\Delta\Delta G_{bind,HET\ AB}$ and $\Delta\Delta G_{bind,HM\ AA}$. (A–D) $\Delta\Delta G_{bind,HM\ AA}$ and $\Delta\Delta G_{bind,HM\ BB}$ remain constant throughout. Changes in the y axis indicate changes in $\Delta\Delta G_{bind,AB}$, with all three values ($\Delta\Delta G_{bind,HM\ AA}$, $\Delta\Delta G_{bind,HET\ AB}$, $\Delta\Delta G_{bind,HM\ BB}$) being identical at zero. (E) Parametric simulations using differences in the specific activities of homo- and heterodimers and varying mean effects of mutations on homomers. HET activity bias indicates a percentage decrease (negative values) or increase (positive values) of the heterodimer activity with respect to the homodimer activity. (F) Percentages of heterodimers and each of the two homodimers for each of the 104 tested structures when simulations consider differences in specific activity between homo- and heterodimers. The two homodimers were distinguished based on their abundance, such that high HM referred to the one with the higher concentration and low HM to the one with a lower concentration.

80 h⁻¹ (Fig. 5E; Appendix Fig. S10). We observed a similar pattern in our simulations with structures (Fig. 5F). Large differences in specific activity lead to 100% homodimers or 100% heterodimers, depending on the complex with higher specific activity. However, when these differences became very small, the system seemed slightly less efficient at eliminating the heterodimers. When the heterodimer was 10% less active than the homodimer, many structures still maintained a small fraction of heterodimers, whereas homodimers were essentially eliminated when the heterodimer was 10% more active. Interestingly, moving the optimal total activity from 80 h⁻¹ to 60 h⁻¹ led to important changes in the fitness landscape (Fig. EV5A) that altered drastically the results of the parametric simulations (Fig. EV5B). Whereas extreme biases in specific activity predictably led to the corresponding enrichment of homodimers or heterodimers depending on the more active dimer, HET activity values between −20% and 20% yielded more complex patterns. In this range of HET activity bias values, the simulations can approach the optimum activity value of 60 h⁻¹ either by keeping a low total concentration of highly active complexes or by keeping a high total concentration of less active complexes, with the likelihood of each scenario depending on the distribution of mutational effects. Indeed, we saw both scenarios in our simulations with single structures (Fig. EV5C). However, when looking at the global trends for our simulations with structures the heteromer was still enriched even when it was around 20% less active than the homodimers (Fig. EV5D) since their distributions have more destabilizing effects on the homomers (see white triangle in Fig. 2B) and slight mutational biases favoring the heterodimer (Fig. 3). Thus, differences in specific activity can easily revert the bias toward heterodimers, but the most likely outcome would still be affected by the underlying distribution of mutational effects and the fitness function.

## Discussion

There are many potential fates for duplicate genes. These often involve divergence of the paralogs in terms of function, regulation, and interactions with other proteins (Conant and Wolfe, 2008). Much work has been dedicated to describing the different outcomes (Hochberg et al, 2018; Marchant et al, 2019; Mallik and Tawfik, 2020) and developing models of the fate of duplicate genes (Force et al, 1999; Lynch and Conery, 2000; Gout and Lynch, 2015; Johri et al, 2022). However, we lack a thorough understanding of the underlying molecular determinants and evolutionary forces. The way mutations themselves can drive evolution through their biases towards specific phenotypic outcomes could play a role. For instance, subfunctionalization is more likely to occur than neofunctionalization because loss-of-function mutations are more likely than beneficial gains of function (Lynch and Force, 2000; Gibson and Goldberg, 2009). This alone could naturally bias the maintenance of gene duplicates in one direction, even in the absence of natural selection favoring one outcome or the other. Here, we examine a special case of subfunctionalization which is the potential replacement of a homodimer by a heterodimer following gene duplication.

We characterized the equilibrium between homo- and heterodimers after duplication. Small effects on binding affinity in the range of 0.5–1 kcal/mol can drastically alter the proportions of homo- and heterodimers. Since our estimations of the distributions of mutational effects are centered close to that range (averages of 0.2 kcal/mol for heterodimers and 0.4 for homodimers), mutations that could shift the equilibrium are likely to be sampled often by evolution. Furthermore, mutational effects on binding for homodimers and heterodimers are highly correlated. As a result, an avenue for removing one of the dimers could be the initial transient destabilization of both followed by the subsequent restabilization of the favored interaction, as observed for duplicated proteins forming heteromers with a common partner (Teufel et al, 2019). However, our FoldX predictions suggest that there is a small percentage of mutations that could directly stabilize one complex while destabilizing the other (Fig. 3A; Fig. EV1A). Our model shows that changes in the proportion of the three complexes could therefore arise through a few mutations, matching previous reports of homomer evolution (Ashenberg et al, 2011; Garcia-Seisdedos et al, 2017; Hochberg et al, 2018) as well as heteromer evolution (Stutz and Blein, 2020; Emlaw et al, 2021).

In our simulated evolutionary trajectories, we observe that the most common outcome is for the proportion of heterodimers to increase. This result extends our previous observation that heterodimers of paralogs are often retained when only the homodimers are selected for (Marchant et al, 2019). However, a key difference is that in our previous study we considered selection acting on the ΔG_bind of one of the complexes, whereas here we consider selection acting on the total activity of the system. As such, the relative concentrations of homo- and heterodimers are allowed to drift as long as the total activity is maintained. The main implication of our results is that heterodimers of paralogs are likely to become the major functional unit over the ancestral homodimeric protein unless there is selection for the homodimer or against the heterodimer. Our conclusions are striking, especially when considering that interactions between identical chains (i.e., homodimers) are more likely to sample highly favorable binding energies when compared to interactions between different chains (i.e., heterodimers) (Lukatsky et al, 2007; André et al, 2008). However, the same authors also identified that real structures of superfamily heterodimers tend to have more favorable contacts between charged residues than homodimers (Lukatsky et al, 2007). The mutational biases we observe would have a similar effect in our simulations to such increases in favorable interactions, even though the increase in the concentration of heterodimers was not explicitly selected for. Interestingly, others have reported that symmetric assemblies tend to dominate over asymmetric ones (André et al, 2008). In the case of heteromers of paralogs, they inherit the global structural symmetry of their ancestral homomers because they come from a gene duplication. However, because the subunits of the heterodimer of paralogs are encoded by different genes, they allow for local differences in sequence that can provide more favorable contacts. Overall, the dominance of heteromers in our simulations is in agreement with previous analyses of the homo and heteromeric fates of paralogs in high-throughput PPI data (Mallik and Tawfik, 2020). However, high-throughput PPI data are typically derived without considering barriers against heteromerization (such as differential compartmentalization and expression) and the relative abundance of each interaction. As such, further experimental characterization of the fates of paralogs would require methods that address these issues.

The case for selection against heteromers might differ among protein families. For example, selection against heteromers of paralogs has been pointed out as a way to reduce cross-talk and promote specialization of transcription factors (Amoutzias et al, 2008), kinases (Ashenberg et al, 2011), and chaperones (Pareek et al, 2011). However, differences in specificity in transcription factors can also be achieved if one of the homomers and the heteromer are retained (Mamnun et al, 2002; Baker et al, 2013). Others have reported positive selection for heteromerization of transcription factors (Hernández-Hernández et al, 2007), as well as allosteric (Kafková et al, 2017) and cooperative (Pillai et al, 2020) contributions of heteromerization, which would act to preserve the heteromer. The efficiency of selection for or against heterodimers might also change over time, for example, once the two paralogs diverge in function or in expression patterns (Kaltenegger et al, 2013). Interestingly, Pils and Schultz, 2004 have suggested that catalytically inactive protein domains are common (between 10% and 15%) in eukaryotic proteomes. These catalytically inactive proteins are often conserved and seem to have evolved regulatory functions mediated by binding to their paralogs (Pils and Schultz, 2004; Adrain and Freeman, 2012), providing a niche to preserve these heteromers. A final consideration about the difficulty of removing heteromers is that the starting concentrations immediately after duplication become more skewed towards heteromers as the number of subunits in the oligomer increases. Because of this skew, higher-order complexes require a greater difference in the binding affinities of homomers and heteromers to promote specificity for homomerization (Hochberg et al, 2018). Thus, it is unclear whether there would be selection against most heteromers of paralogs and if it could be efficient enough to eliminate them.

Our results add to a growing body of work suggesting that complexity can increase neutrally (Stoltzfus, 1999; Lynch, 2007; Hochberg et al, 2020; Muñoz-Gómez et al, 2021; Schulz et al, 2022; Abrusán and Foguet, 2023). In the case of heteromers of paralogs, two proteins would become necessary to provide the function of an ancestral homomeric protein. Such transitions increase the complexity of protein interaction networks, and they do not necessarily provide an immediate adaptive benefit. A feature that can only be introduced by heteromers is mutational asymmetry, which allows distinct sequences for the different subunits. Mutants that would otherwise eliminate catalytic activity in a homomeric assembly can participate in heteromers with native-like activity (Ebert et al, 2015). Indeed, the distributions we obtained when simulating mutational effects in silico had smaller effects on heterodimers than on homodimers and often had small enrichments in heterodimer-favoring mutations (Fig. 3E,F). As a result, their fixation could be a case of "survival of the flattest" due to the smaller effects of point mutations (Wilke et al, 2001). However, this comes with increases in network complexity that do not contribute to robustness (Diss et al, 2017; Dandage and Landry, 2019). Overall, non-adaptive processes play a significant role in the abundance of heteromers of paralogs in extant protein interaction networks, although there are known examples of heteromers that provide an adaptive benefit (Bridgham et al, 2008; Pillai et al, 2020). Future work should focus on experimental characterizations of the distribution of mutational effects of protein structures. While our analyses point to the existence and effects of mutational biases, we could not fully establish their structural determinants. Comparing

larger-scale datasets might allow isolating the effects of individual structural features on the distribution of mutational effects.

Importantly, our model makes multiple simplifying assumptions. In order to keep the simulations scalable, the distribution of mutational effects on folding free energy and binding affinity is kept constant throughout, instead of being recalculated at every step. We make this assumption also because while it is well-known that the effects of each new mutation are contingent on previous mutations, whether the overall shape of the distribution stays the same is unclear. Similarly, any systematic errors in our FoldX predictions would propagate as we simulate more mutations, making the estimations progressively less accurate. However, given that we consider the heterogeneity of the effects of mutations among the starting complexes, it is likely correct to assume that the distribution of effects of mutations through time is captured in this initial heterogeneity. Finally, our main conclusion about the likelihood of heterodimers to become more abundant than homodimers depends on the shapes of the FoldX-derived distributions of mutational effects. As shown in Fig. 2B, modifying these distributions would lead to different results. Thus, generating experimental data about how mutations affect the binding affinities of homodimers and heterodimers of paralogs could help address this limitation.

We do not consider the potential effects of other covariates that would prevent interactions between paralogs, such as compartmentalization (Marchant et al, 2019) and cotranslational assembly (Shiber et al, 2018; Badonyi and Marsh, 2023). In these cases, the interaction between the two paralogs would not be possible, and the system would be expected to preserve the homomers. Our model also does not consider that tight binding could provide cross-stabilizing contributions to protein folding, such as the ones modeled in Rotem et al, (2018), although the fraction of properly folded subunits in our simulations consistently stays above 90% in most of our simulations. Finally, as mutations accumulate, one might expect the specific activity of homomers and heteromers to evolve. A special case of this evolution could be the emergence of dominant negative mutations, which would inactivate the homodimer of the mutated paralog and the heterodimer (Veitia, 2007). Depending on the optimal activity, this reduction in activity could either lead to selection against the heterodimer to restore activity or lead to the emergence of new regulatory roles for the heterodimer (Bridgham et al, 2008). A second special case is that of loss-of-function mutations that inactivate the homomers but yield active heteromers (Després et al, 2024). This scenario would be similar to the ones explored here, in which the specific activity of the homodimers is greatly decreased with respect to that of the heterodimer. However, the distribution of mutational effects on specific activity would be unique for each protein and without more knowledge on the distribution of mutational effects on specific functions, we could make arbitrary choices that could bring the balance in one direction or another. Relaxing our assumptions and including more cases would allow our model to more accurately represent the fate of the paralogs, but our model already shows the complex interplay between synthesis rates, folding energies, binding affinity, and specific activities.

Our results highlight the importance of exploring neutral hypotheses for the evolution of molecular systems. While clearly natural selection plays a role in shaping them, there appears to be

an important role of neutral evolution and mutational biases as well. Our result that heteromers of paralogs are likely to replace ancestral homomers after duplication agrees with recent observations that heteromers are more abundant in eukaryotic proteomes than in prokaryotes (Mallik and Tawfik, 2020; Schweke et al, 2024). This is consistent with the higher population sizes of prokaryotes (Lynch, 2006; Charlesworth, 2009), since any kind of selection against heteromers would be more efficient in these species than in eukaryotes. Future work should continue to investigate the factors leading to the fate of paralogs, focusing on how both natural selection and neutral evolution have contributed to the modern levels of complexity of protein interaction networks.

# Methods

## Reagents and tools table

| Software | | |
|---|---|---|
| FoldX 5.0 | https://foldxsuite.crg.eu/ | Delgado et al, 2019. *Bioinformatics* |
| MutateX 1.0 | https://github.com/ELELAB/mutatex | Tiberti et al, 2022. *Brief Bioinform.* |
| DSSP 2.2.1 | https://swift.cmbi.umcn.nl/gv/dssp/ | Kabsch and Sander, 1983. *Biopolymers* |
| CD-Hit 4.8.1 | https://sites.google.com/view/cd-hit | Li and Godzik, 2006. *Bioinformatics*; Fu, et al, 2012. *Bioinformatics.* |
| Python 3.8 | https://www.python.org/downloads/ | van Rossum and Drake, 2009. |
| **Databases** | | |
| Evolutionary Classification of Protein Domains (ECOD) | http://prodata.swmed.edu/ecod/ | Cheng et al, 2014. *PLoS Comput. Biol.* |
| Protein Data Bank (PDB) | https://www.rcsb.org/ | Berman et al, 2000. *Nuc. Ac. Res.* |

## Methods and protocols

### Protein structures
#### Selection of protein structures

The complete set of structures deposited on the Protein Data Bank (PDB) (Berman et al, 2000) were downloaded on March 29th, 2021. Structures were then filtered based on the following criteria:

- Experimental data: X-ray crystallography
- Author-assigned biological assembly: Dimeric
- Structures must have two identical subunits (no missing residues or both subunits miss the same residues)
- Sequence length was restricted to 60–450 residues.

We mapped the remaining set of structures to their ECOD architecture annotations (Cheng et al, 2014). We selected structures from the major architecture types such that we had at least seven of each one. Our final set comprised 104 PDB structures. These structures were clustered with CD-Hit version 4.8.1 (Li and Godzik,

2006; Fu et al, 2012) with the following parameters: -n 2 (word size), -c 0.4 (sequence identity threshold), -G 1 (global alignment), and -s 0.8 (length difference cutoff). All of our selected structures were identified in different clusters, confirming low sequence identity between them.

#### Structural analyses

We studied different structural parameters for each protein structure (Appendix Fig. S5). Secondary structures were annotated with DSSP version 2.2.1 (Kabsch and Sander, 1983). Interfaces were identified following Tsai et al, (1996). Residues were considered to be at the interface core if the distance between one of their non-hydrogen atoms and a non-hydrogen atom from any residue of the other subunit was smaller than the sum of their van der Waals radii plus 0.5 Å. Residues within 6 Å of the interface core but not in contact with the other subunit were classified as the interface rim. We analyzed the proportion of residues in contact with their counterparts from the other subunit as a proxy for symmetry at the interface. For this analysis, we used a cutoff distance of 4 Å between any non-hydrogen atoms of the two residues on the two chains.

## Estimation of the distribution of mutational effects with FoldX

The biological assemblies of selected PDB structures were generated using custom scripts (Marchant et al, 2019). Energy minimization was performed on the biological assemblies using the FoldX Repair function (Delgado et al, 2019) ten times to ensure convergence (Usmanova et al, 2018). Repaired structures were then used to perform in silico mutagenesis using the FoldX BuildModel and AnalyseComplex functions (Delgado et al, 2019) with the MutateX workflow (Tiberti et al, 2022) to estimate mutational effects on binding affinity and folding free energy. Mutations were simulated in two runs for each complex: one run in which mutations were applied on both subunits (simulating mutational effects for homomers) and one run in which mutations were applied on only one subunit (simulating mutational effects for heteromers). Mutational effects on binding affinity were taken from the corresponding distributions, while mutational effects on folding free energy were taken from the distribution of effects on the heteromer since our model uses changes on folding free energy to calculate the fraction of properly folded subunits available for complex assembly.

### Framework for simulations
#### Pre-duplication model

Our simulations consider a system of a gene encoding a protein (A) that forms a homodimer (AA). Protein copies are synthesized with synthesis rate $s_A$ and typical values for starting parameters of folding free energy ($\Delta G_{fold} = -5$ kcal/mol) (Pace, 1975; Plaxco et al, 2000) and binding affinity ($\Delta G_{bind} = -10$ kcal/mol) (Choi et al, 2015; Jankauskaitė et al, 2018). The proportion of properly folded subunits is calculated based on Eq. (1) (Sailer and Harms, 2017):

$$w = \frac{1}{1 + e^{\Delta G_{fold}}}$$

(1)

where $w$ is the fraction of folded proteins and $\Delta G_{fold}$ is the free energy of the native fold.

Properly folded subunits are considered for the formation of complexes while the remaining misfolded subunits are removed (assumed to be degraded/not to participate in the formation of complexes). Decay rates for properly folded monomers ($d_A$) and homodimers ($d_{AA}$) are set to 1.3 h$^{-1}$, following the derivation by Hausser et al, (2019) based on median protein half-life and cell division rate for yeast. The assembly of homodimers is driven by the association constant $k_{AA}$. Thus, using $s_A$, $w$, $d_A$, $d_{AA}$, and $k_{AA}$ we can calculate the expected concentrations at the equilibrium of A and AA (see full derivation in Appendix Note 1). The equilibrium concentrations of monomers and homodimers are multiplied by their specific activity (0.1 for monomers, 1 for the homodimer) to calculate the total activity of the system. Fitness is determined from a lognormal function (defined by alpha and beta parameters) from the total activity of the system. Since the total activity before duplication was calculated at 38.2 h$^{-1}$, the optimum fitness was set at 60 h$^{-1}$ or 80 h$^{-1}$ (alpha parameter) to favor duplication events. The beta parameter was set to 0.5 so that fitness would be reduced by half when the total activity was either half or twice the alpha parameter. All simulations were written in custom Python 3.8 (Van Rossum and Drake, 2009) scripts.

At the start of the simulation, each mutation has a probability of being a mutation on the coding sequence, a duplication, or a mutation causing a change in the synthesis rate. The effects of coding mutations on folding free energy ($\Delta\Delta G_{fold}$) and binding affinity ($\Delta\Delta G_{bind}$) are sampled from distributions of mutational effects (see section "Parametric distributions of mutational effects" below) and are used to calculate the fraction of properly folded subunits and the equilibrium constants, respectively. Equation (2) shows how the association constants for each dimer are calculated once $\Delta G_{bind}$ is updated with the effect of the sampled mutations ($\Delta\Delta G_{bind}$):

$$K_{AA} = e^{\frac{-\left(\Delta G_{bind,AA} + \Delta\Delta G_{bind,AA}\right)}{RT}} \tag{2}$$

where $R$ is the gas constant ($1.987 \times 10^{-3}$ cal K$^{-1}$ mol$^{-1}$) and $T$ is the temperature at 298 K.

Once a mutation has been sampled, the equilibrium concentrations of all molecular species and the total activity of the system are recalculated using Eqs. (3) and (4) (full derivation in Appendix Note 1):

$$c_A = \frac{-d_A + \sqrt{d_A^2 + 8d_{AA}k_{AA}s_A}}{4d_{AA}k_{AA}} \tag{3}$$

$$c_{AA} = k_{AA}c_A^2 \tag{4}$$

where $c_A$ and the $c_{AA}$ are the concentrations of each molecular species, $d_A$ and $d_{AA}$ are the decay rates of each molecular species, $s_A$ is the rate of synthesis of properly folded subunits of A, and $k_{AA}$ is the association constant for AA.

Using the new equilibrium concentrations, the fitness of the new state is compared to that of the previous state. Simulations continue until 200 mutations have been fixed, with the first one being the duplication. Duplications introduce new molecular species: monomer B, homodimer BB, and heterodimer AB. As a result, the

calculations for the equilibrium concentration of each complex change (see the next section).

*Post-duplication model*

Once a duplication has been fixed, the system will have two genes encoding two proteins (A and B) that will assemble into complexes AA, AB, and BB. Mutations continue to be sampled from the underlying distribution of mutational effects, and the system no longer samples for further duplication events. Once a mutational effect is selected, the affected protein (A or B) is selected at random. Afterwards, the new $\Delta G_{fold}$ is calculated to estimate the fraction of properly folded subunits and the $\Delta G_{bind}$ for both the homodimeric and heterodimeric interactions are used to estimate the new equilibrium constants. Following Hochberg et al, (2018), a factor of 2 is used to correct for the mixing entropy difference that inherently favors heteromers in the mixture of heterodimers and homodimers. Once all the values are calculated, the 4th-degree polynomial in Eq. (5) is used to estimate the concentration at the equilibrium of A (full derivation in Appendix Note 2):

$$0 =$$
$$2d_{AA}k_{AA}\left(4\frac{d_{AA}k_{AA}d_{BB}k_{BB}}{d_{AB}k_{AB}} - d_{AB}k_{AB}\right)c_A^4$$
$$+$$
$$\left(d_A\left(8\frac{d_{AA}k_{AA}d_{BB}k_{BB}}{d_{AB}k_{AB}} - d_{AB}k_{AB}\right) - 2d_Bd_{AA}k_{AA}\right)c_A^3$$
$$+$$
$$\left(\frac{2d_{BB}k_{BB}d_A^2}{d_{AB}k_{AB}} + s_A\left(d_{AB}k_{AB} - 8\frac{d_{AA}k_{AA}d_{BB}k_{BB}}{d_{AB}k_{AB}}\right) - d_Ad_B - d_{AB}k_{AB}s_B\right)c_A^2$$
$$+$$
$$s_A\left(d_B - 4d_A\frac{d_{BB}k_{BB}}{d_{AB}k_{AB}}\right)c_A$$
$$+$$
$$2s_A^2\frac{d_{BB}k_{BB}}{d_{AB}k_{AB}}$$

$$\tag{5}$$

where $dA$, $dB$, $dAA$, $dAB$, and $dBB$ are decay rates for each molecular species; $sA$ and $sB$ are the synthesis rates of properly folded subunits of A and B; $kAA$, $kAB$, and $kBB$ are the association constants for each of the dimers.

Once the four potential solutions to the concentration of A are known, the concentrations of the rest of the molecular species are calculated using Eqs. (6–9):

$$c_B = \frac{s_A/c_A - 2d_{AA}k_{AA}c_A - d_A}{d_{AB}k_{AB}} \tag{6}$$

$$c_{AA} = k_{AA}c_A^2 = \frac{r_2}{d_{AA} + r_1}c_A^2 \tag{7}$$

$$c_{BB} = k_{BB}c_B^2 = \frac{r_4}{d_{BB} + r_3}c_B^2 \tag{8}$$

$$c_{AB} = k_{AB}c_Ac_B = \frac{r_6}{d_{AB} + r_5}c_Ac_B \tag{9}$$

where $r_1$, $r_3$, and $r_5$ are dissociation rates of AA, BB, and AB, respectively; $r_2$, $r_4$, and $r_6$ are association rates of AA, BB, and AB, respectively.

Among the four possible solutions to the complete system of equations, the physically correct solution is the one that results in positive values for the concentrations of all five molecular species.

*Parametric distributions of mutational effects*

We performed simulations by sampling from multivariate normal distributions of mutational effects. Each multivariate normal distribution was used to sample effects on $\Delta G_{fold}$ for each monomer, $\Delta G_{bind, AA}$ for the homodimer, and $\Delta G_{bind, AB}$ for the heteromer. We set the means of the multivariate normal distributions and the correlations between variables to values similar to those observed in the distributions simulated with FoldX:

- Effects on $\Delta G_{fold}$: $N(2.6, 4.6)$
- Effects on $\Delta G_{bind, AB}$: $N(0.2, 1.2)$
- Effects on $\Delta G_{bind, AA}$: The mean was allowed to vary between 0.06 and 0.6, while the standard deviation varied between 0.6 and 3 (Fig. 2B). For all other figures showing results of simulations with parametric distributions, the standard deviation was kept constant at 2.4.
- Correlation between effects on $\Delta G_{fold}$ and $\Delta G_{bind, AB}$: 0.3
- Correlation between effects on $\Delta G_{fold}$ and $\Delta G_{bind, AA}$: 0.3
- Correlation between effects on $\Delta G_{bind, AA}$ and $\Delta G_{bind, AB}$: 0.9.

#### Simulations allowing changes in gene expression

A subset of the simulations were run, allowing mutations to have an effect on gene expression. For each sampled mutation, a given probability (ranging from 0.1 to 0.9) was given that a mutation would have an effect on synthesis rates instead of the coding sequence. The underlying distribution of mutational effects on gene expression was based on a fit to previously published work on the TDH3 promoter (Metzger et al, 2016). The data were fit to a skew-normal distribution, which gave the following parameters: mean = 0, standard deviation = 0.025, skew = −0.125. Since the mutational effects reported by Metzger et al, (2016) were normalized with respect to the WT, they were applied by multiplying the sampled values times the current synthesis rate.

*Simulations with biases in specific activity*

A subset of the simulations were run to explore the outcomes when the homo- and heterodimers had different specific activities. To bias a simulation towards homodimers, the specific activity for homodimers was kept at 1 while the specific activity of heterodimers was given a value ranging from 0.1 to 0.9, and vice versa to bias simulations toward heterodimers.

#### Data analysis

For each structure, we analyzed the results of 50 replicate simulations. Each replicate continued until 200 mutations were fixed. We tracked the equilibrium concentrations of all molecular species (monomers A and B; homodimers AA and AB; and heterodimer BB) through time. We defined terms for the relative concentration of each molecular species at the end of the simulation according to Eq. (10):

$$p_X = \frac{100 * c_X}{c_A + c_B + c_{AA} + c_{AB} + c_{BB}} \quad (10)$$

where $pX$ indicates the relative concentration of X (X being any of A, B, AA, AB, or BB) and $cX$ indicates the concentration of X.

We analyzed enrichments in homomers or heteromers by looking for deviations from the initial conditions post duplication (25% AA, 50% AB, 25% BB). As such, replicates were classified based on the relative concentrations of each species at the end of the simulation:

- HET dominant: 70 <= pAB
- HM dominant: 70 <= (pAA + pBB)
- Both HM and HET: 70 <= (pAB + pAA + pBB) AND 70 >= pAB AND 70 >= (pAA + pBB)
- Monomers: 70 <= (pA + pB)
- Ambiguous: 70 >= (pAB + pAA + pBB) AND 70 >= (pA + pB).

Our measure for the results of the simulation was the concentration of each molecular species at the end, averaged over all replicates.

## Data availability

The datasets and computer code produced in this study are available in the following databases: Scripts: Github (https://github.com/Landrylab/Homomer_duplication_2024) [https://doi.org/10.5281/zenodo.10659566]. Supporting datasets: Zenodo (https://zenodo.org/records/10048861) [https://doi.org/10.5281/zenodo.10048861].

## Peer review information

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

## Acknowledgements

The authors thank François Rouleau, Soham Dibyachintan, Adrian Serohijos, Sarah Otto, and members of the Landry and Levy labs for their feedback and help on the project. The computer cartoon used to denote parametric simulations was created by Jan Flessau. This work was supported by Canadian Institutes of Health Research (CIHR) Foundation grant 387697 (to CRL), Canada Research Chair in Cellular Systems and Synthetic Biology (to CRL), the Fonds de recherche du Québec—Nature et technologies (FRQNT) (Merit Scholarship Program for Foreign Students, dossier 290237, to AFC), joint funding from MEES and AMEXCID (to AFC), Mitacs Globalink Research Award (award number IT28316, to AFC), Alexander Graham Bell fellowship from the National Sciences and Engineering Research Council of Canada (NSERC) (to LN-T), FRQNT funding (to LN-T), European Research Council (ERC) under the European Union's Horizon 2020 research and innovation program (grant agreement No. 819318, to EDL), by the Israel Science Foundation grant no. 1452/18 (to EDL), and by the Abisch-Frenkel Foundation (to EDL).

## Author contributions

**Angel F Cisneros**: Conceptualization; Data curation; Software; Formal analysis; Funding acquisition; Validation; Investigation; Visualization; Methodology; Writing—original draft; Writing—review and editing. **Lou Nielly-Thibault**: Conceptualization; Data curation; Software; Investigation; Methodology; Writing—review and editing. **Saurav Mallik**: Validation; Investigation; Writing —review and editing. **Emmanuel D Levy**: Supervision; Validation; Investigation; Writing—review and editing. **Christian R Landry**: Conceptualization; Supervision; Funding acquisition; Investigation; Methodology; Writing—original draft; Project administration.

## Disclosure and competing interests statement

The authors declare no competing interests. Christian Landry is a member of the Advisory Editorial Board of *Molecular Systems Biology*. This has no bearing on the editorial consideration of this article for publication.

# Expanded View Figures

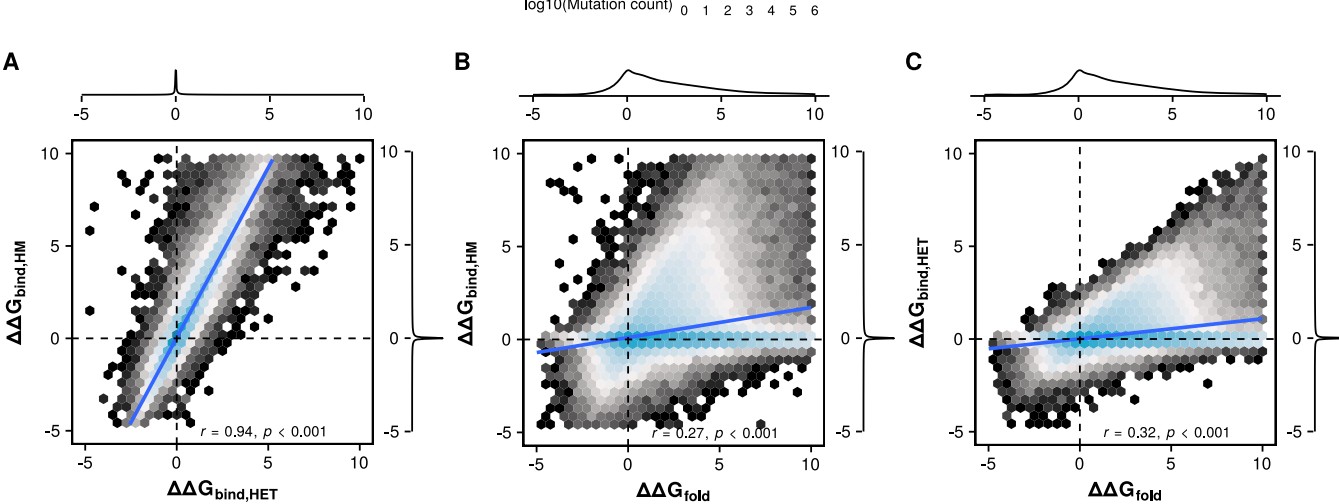

**Figure EV1.  Bidimensional distributions of mutational effects on $\Delta G_{bind,HET}$, $\Delta G_{bind,HM}$, and $\Delta G_{fold}$.**

(A–C) Hexagonal bins indicate the density of mutational effects for the pool of mutations for all structures. Two variables and their correlations are shown in each panel: $\Delta G_{bind,HET}$ and $\Delta G_{bind,HM}$ (A), $\Delta G_{bind,HM}$ and $\Delta G_{fold}$ (B), $\Delta G_{bind,HET}$ and $\Delta G_{fold}$ (C). *P* values are calculated using an asymptotic confidence interval based on Fisher's Z transform for Pearson correlation coefficients.

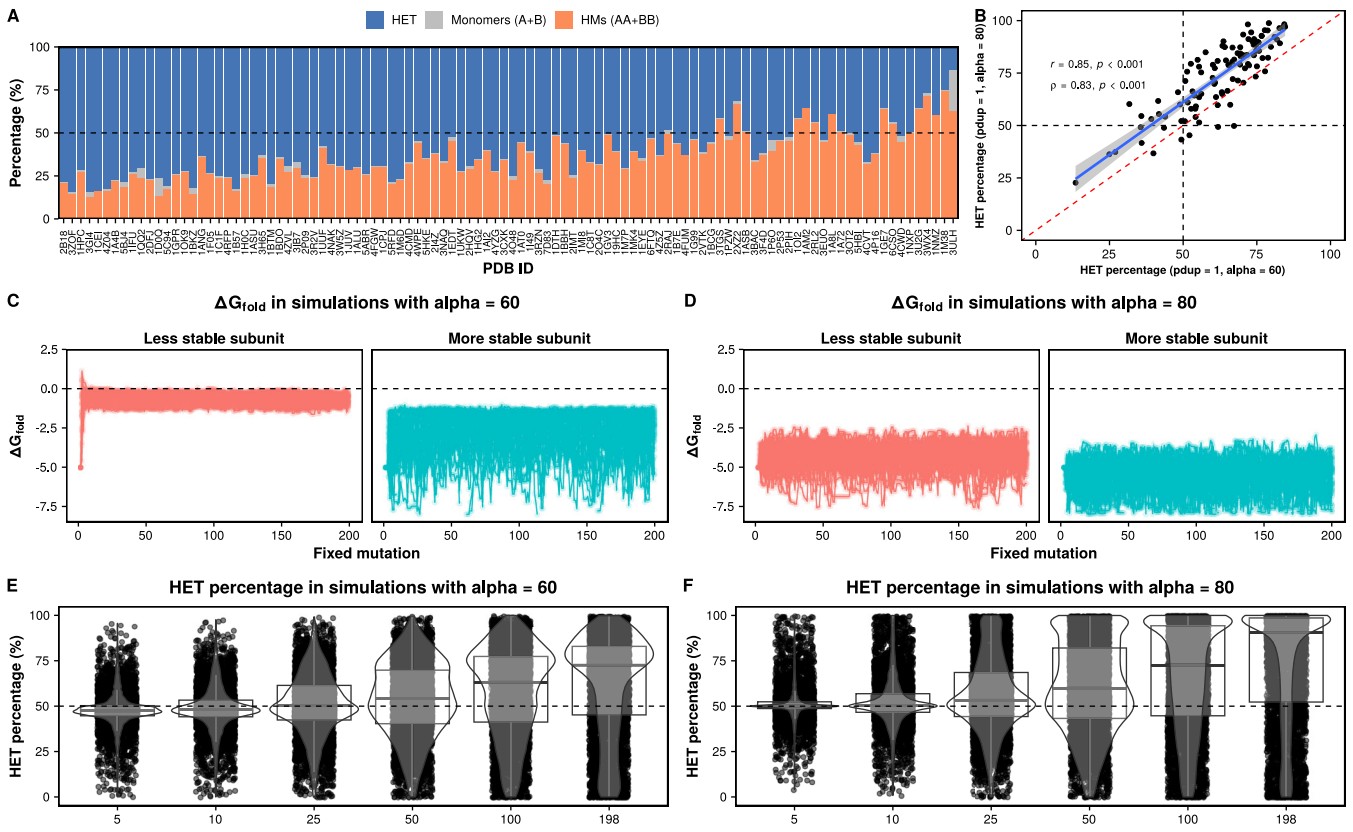

**Figure EV2. The general bias observed for heterodimers is also observed when the optimal activity is set to a slightly lower value (alpha = 60).**

(A) Percentage of heterodimers, homomers, and monomers at the end of simulations with distributions of mutational effects structures when the optimal activity (alpha) is set to 60. Structures are listed in the same order as in Fig. 2C. (B) Correlation between the percentages of heterodimers observed at the end of simulations with alpha = 60 (data from Fig. S6A) and simulations with alpha = 80 (data from Fig. 2C). The *P* value for the Spearman correlation coefficient in panel B is calculated using the asymptotic *t* approximation method. The P value for the Pearson correlation coefficient is calculated using an asymptotic confidence interval based on Fisher's Z transform. (C, D) Evolution of $\Delta G_{fold}$ values in simulations with alpha = 60 (C) and alpha = 80 (D). For simplicity, only the data for PDB: 1GPR are shown. (E, F) Evolution of the percentage of heteromers in simulations with alpha = 60 (E) and alpha = 80 (F). Replicates for all simulations with all structures are shown. Boxplots in (E, F) indicate the median (center lines) and interquartile range (hinges). Whiskers extend from the hinges of each box to the most extreme values that are at most 1.5 times the interquartile range away from the hinges.

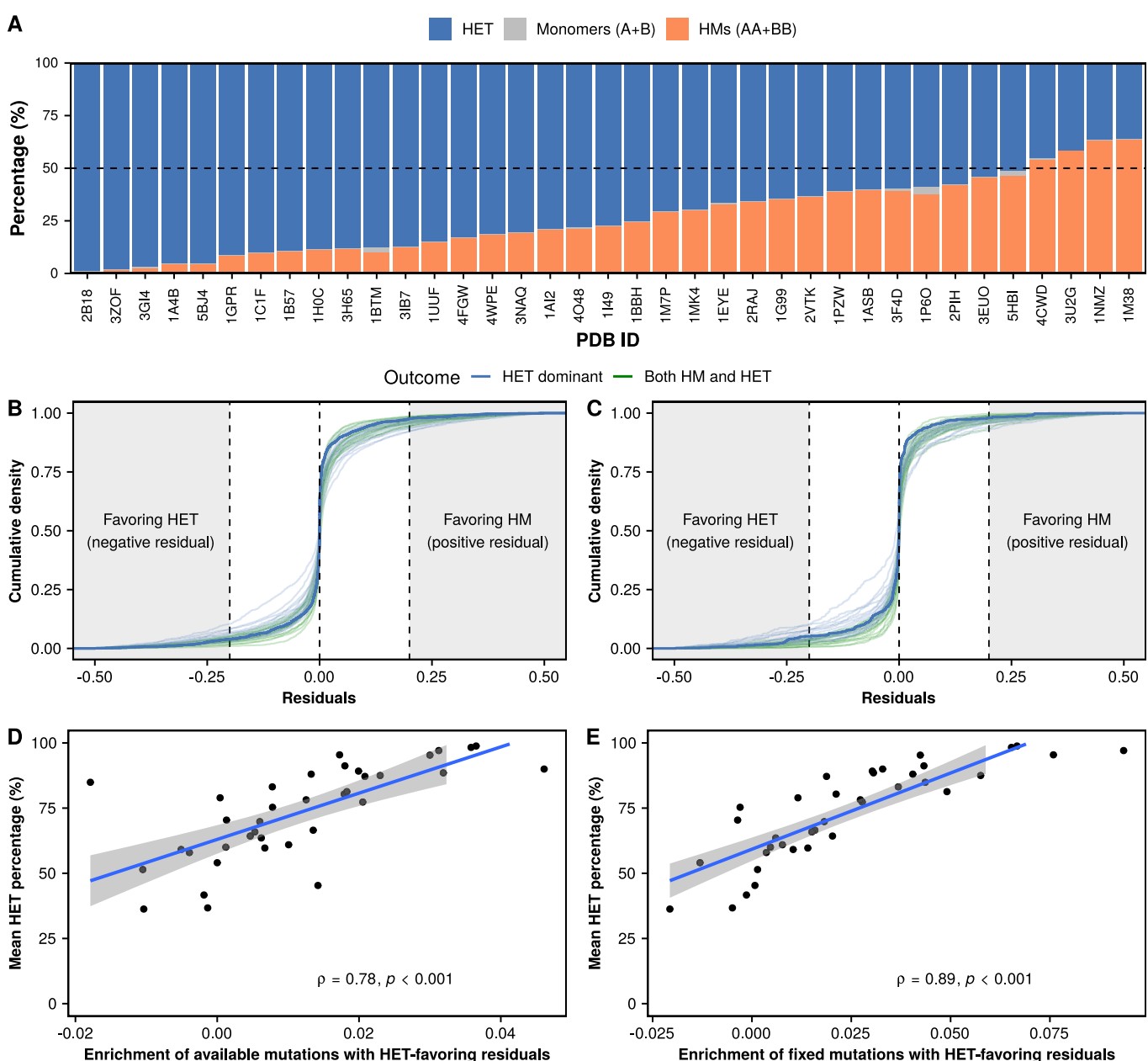

**Figure EV3. Mutational biases promote the enrichment of heterodimers in very high quality PDB structures.**

(**A**) Average percentages of heterodimers, homodimers, and monomers for each of the very high quality PDB structures at the end of the simulations. The dashed line indicates the starting point at 50% heterodimers and 50% homodimers (25% of each homodimer). (**B, C**) Distributions of available (**B**) and fixed (**C**) residuals with respect to the expectation for each of the 37 very high quality structures. Colors indicate mean outcomes for each structure: HET dominant (heterodimer >70% of dimers), HM dominant (homodimer >70% of dimers), both HM and HET (rest of cases). The thicker blue line represents 2B18. Shaded regions on the left and right indicate mutations whose residuals have a magnitude of at least 0.2 favoring either the heterodimer (negative values) or the homodimer (positive values). (**D, E**) Correlation between enrichment of available (**D**) and fixed (**E**) mutations with residuals favoring the heterodimer and the mean percentage of heterodimers at the end of the simulations. The enrichment of heterodimer-favoring residuals is calculated as the density of mutations with residuals smaller than −0.2 minus the density of mutations with residuals greater than 0.2. *P* values for (**D, E**) are calculated using the asymptotic t approximation method for Spearman correlation coefficients.

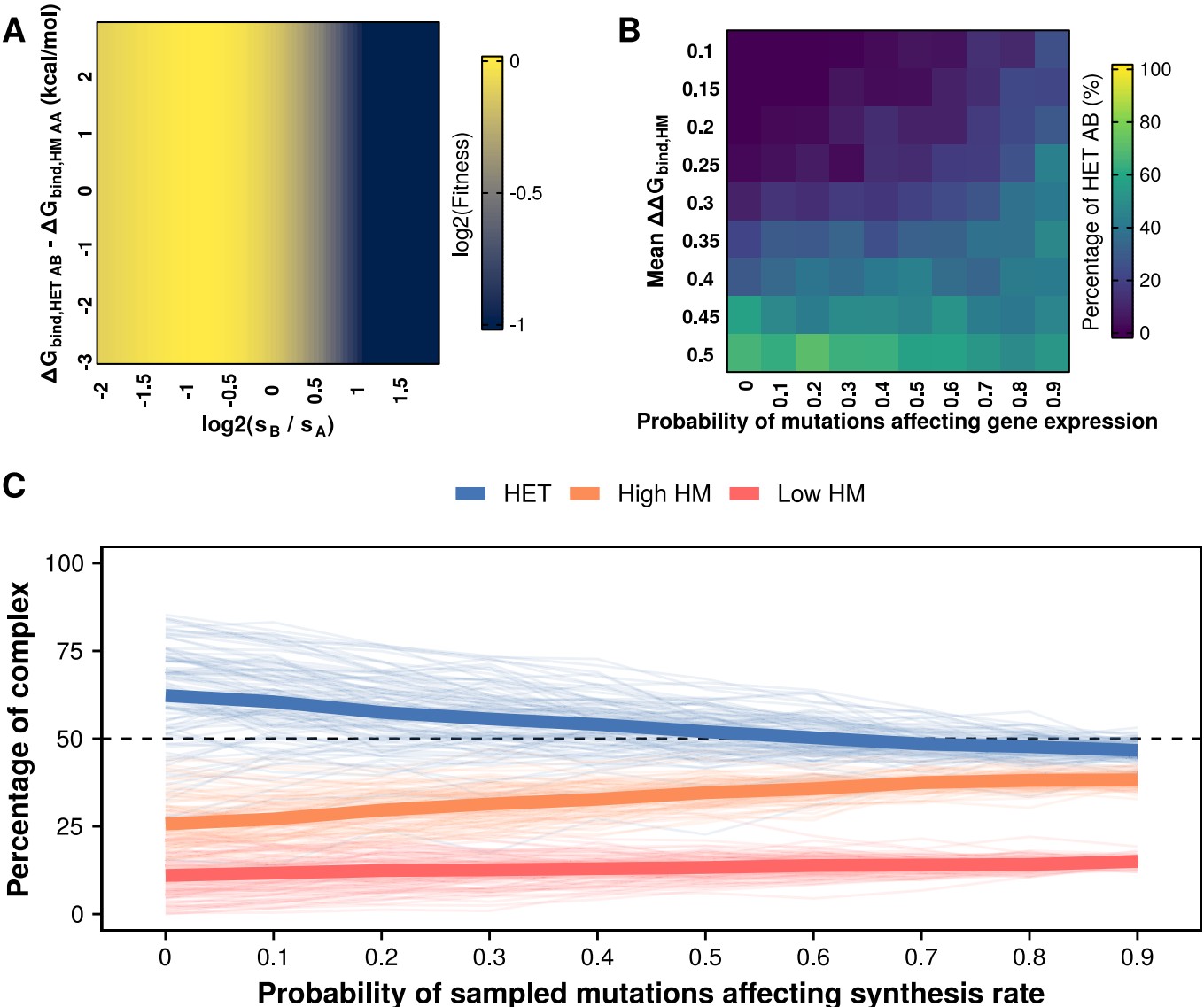

**Figure EV4. Simulations allowing changes in synthesis rates with lower optimal activity (alpha = 60).**

(A) Fitness landscape when the optimal activity is set to 60. Note that $s_A$, $\Delta\Delta G_{bind,AA}$, and $\Delta\Delta G_{bind,BB}$ remain constant throughout. (B) Percentages of heterodimers at the end of parametric simulations allowing changes in synthesis rates and varying mean effects of mutations on homodimers. (C) Percentages of heterodimers and each of the two homodimers for each of the 104 tested structures when simulations consider different probabilities of mutations affecting synthesis rates. The two homodimers were distinguished based on their abundance.

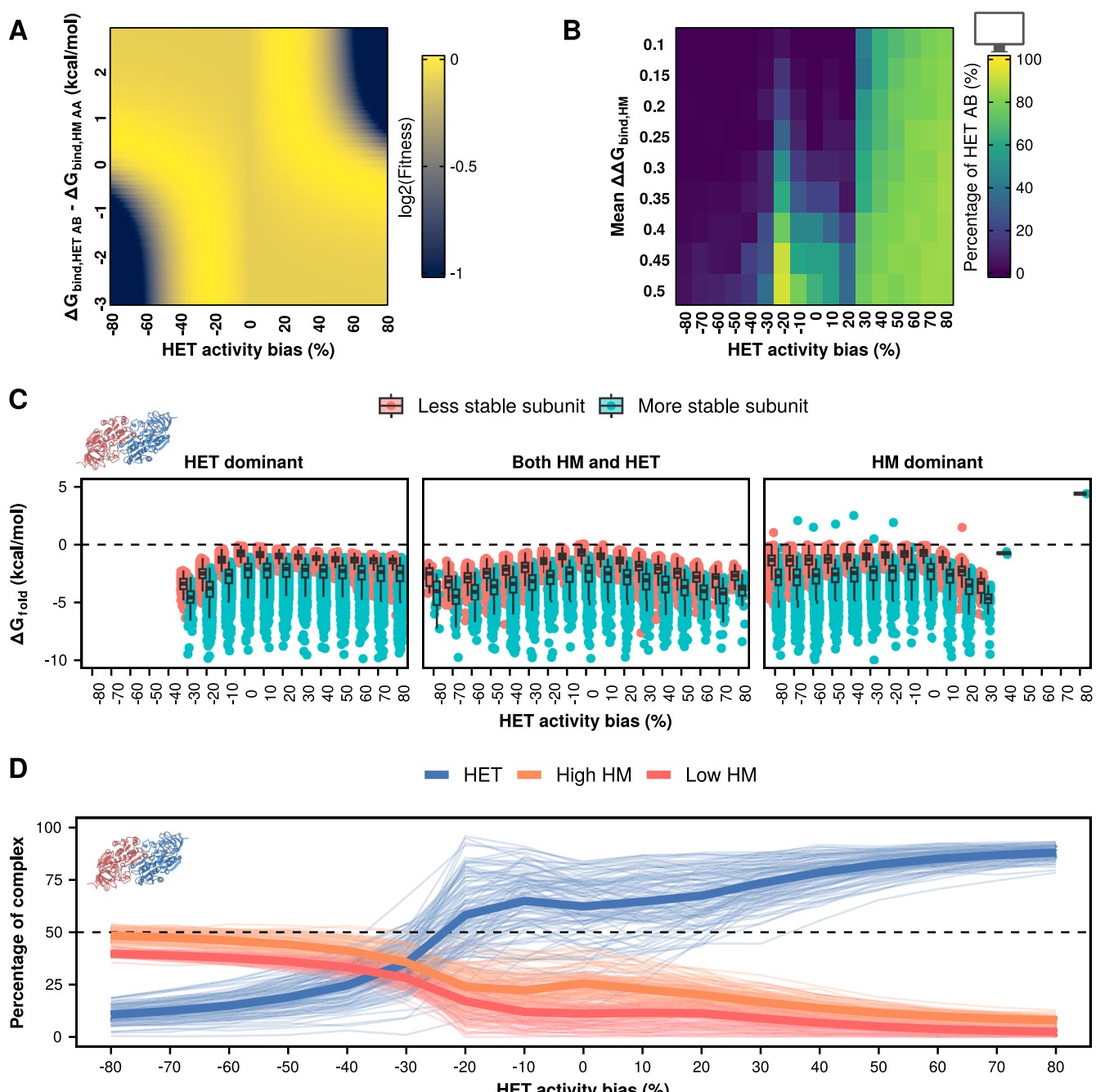

**Figure EV5.  Simulations with differences in specific activity and lower optimal total activity (alpha = 60).**

(A) Fitness landscape when the optimal activity is set to 60. $\Delta\Delta G_{bind,AA}$, and $\Delta\Delta G_{bind,BB}$ remain constant throughout at −10 kcal/mol. (B) Percentages of heterodimers at the end of parametric simulations with different values of specific activity and varying mean effects of mutations on homodimers. (C) $\Delta G_{fold}$ values at the end of simulations with alpha = 60 and different specific activities for homo- and heterodimers. Disfavored outcomes, such as HET dominant when the heterodimer is less active, can still be reached if the subunits remain stable. Boxplots in (C) indicate the median (center lines) and interquartile range (hinges). Whiskers extend from the hinges of each box to the most extreme values that are at most 1.5 times the interquartile range away from the hinges. (D) Percentages of heterodimers and each of the two homodimers for each of the 104 tested structures when simulations consider different specific activities of homo- and heterodimers. The two homodimers were distinguished based on their abundance.

