## [Peer Review File · Molecular Systems Biology]

Mutational biases favor complexity increases in protein interaction networks after gene duplication

Angel Cisneros Caballero, Lou Nielly-Thibault, Saurav Mallik, Emmanuel levy, and Christian Landry

Corresponding author(s): Christian Landry (christian.landry@bio.ulaval.ca)

Review Timeline:

Submission Date:	9th Jan 24
Editorial Decision:	5th Feb 24
Revision Received:	16th Feb 24
Editorial Decision:	23rd Feb 24
Revision Received:	27th Feb 24
Accepted:	28th Feb 24

Editor: Maria Polychronidou

Transaction Report:

5th Feb 2024

Manuscript Number: MSB-2024-12212

Title: Mutational biases favor complexity increases in protein interaction networks after gene duplication

Dear Christian,

Thank you again for submitting your work to Molecular Systems Biology. We have now heard back from the three reviewers who agreed to evaluate your study. As you will see below, the reviewers find the study interesting and are quite supportive. They do however raise a series of (mostly minor) concerns, which we would ask you to address in a revision.

I think that the reviewers' recommendations are clear and seem straightforward to address. I therefore see no need to repeat any of the comments listed below. All issues raised by the referees would need to be satisfactorily addressed. Please let me know in case you would like to discuss in further detail any of the issues raised, I would be happy to schedule a call.

On a more editorial level, we would ask you to address the following points:

- Please provide a .doc version of the manuscript text (including legends for the main figures) and individual production quality figure files for the main Figures (one file per figure).
- We have replaced Supplementary Information by the Expanded View (EV format). In this case, all additional figures and Tables can be included in a PDF called Appendix. Appendix figures and Tables should be labeled and called out as: "Appendix Figure S1, Appendix Figure S2... Appendix Table S1..." etc. Each legend should be below the corresponding Figure/Table in the Appendix. Please include a Table of Contents in the beginning of the Appendix. For detailed instructions regarding expanded view please refer to our Author Guidelines: .
- Supplementary Files 1 and 2 should be included in the Appendix (e.g. as Appendix Notes 1 and 2).
- Table S1 is rather complex, it should be provided as Dataset EV1. Please include a description of the Dataset in a separate sheet in the xls file.
- Table S2 should be provided as Table EV1.
- Please provide a "standfirst text" summarizing the study in one or two sentences (approximately 250 characters), three to four "bullet points" highlighting the main findings and a "synopsis image" (550px width and max 400px height, jpeg format) to highlight the paper on our homepage.
- All Materials and Methods need to be described in the main text. We would encourage you to use 'Structured Methods', our new Materials and Methods format. According to this format, the Material and Methods section should include a Reagents and Tools Table (listing key reagents, experimental models, software and relevant equipment and including their sources and relevant identifiers) followed by a Methods and Protocols section in which we encourage the authors to describe their methods using a step-by-step protocol format with bullet points, to facilitate the adoption of the methodologies across labs. More information on how to adhere to this format as well as downloadable templates (.doc or .xls) for the Reagents and Tools Table can be found in our author guidelines: . An example of a Method paper with Structured Methods can be found here:
- Please include a "Disclosure & Competing Interests Statement".
- Please include a "Data availability" section describing how the data, code etc. have been made available. This section needs to be formatted according to the example below:
The datasets and computer code produced in this study are available in the following databases:
 - Chip-Seq data: Gene Expression Omnibus GSE46748 (<https://www.ncbi.nlm.nih.gov/geo/query/acc.cgi?acc=GSE46748>)
 - Modeling computer scripts: GitHub (<https://github.com/SysBioChalmers/GECKO/releases/tag/v1.0>)
 - [data type]: [full name of the resource] [accession number/identifier] ([doi or URL or identifiers.org/DATABASE:ACCESSION])
- For data quantification: please specify the name of the statistical test used to generate error bars and P values, the number (n) of independent experiments (specify technical or biological replicates) underlying each data point and the test used to calculate p-values in each figure legend. The figure legends should contain a basic description of n, P and the test applied. Graphs must include a description of the bars and the error bars (s.d., s.e.m.).
- Molecular Systems Biology supports formal data citations in the Reference list, to cite previously published datasets. In addition to citing the original papers that reported the data, we encourage you to also cite the relevant datasets directly in the Reference

list. In the text, references to datasets are included as "Data ref: Smith et al, 2001" or "Data ref: NCBI Sequence Read Archive PRJNA342805, 2017". In the Reference list, data citations are very similar to normal literature references but must be labeled with "[DATASET]" at the end of the reference. For detailed instructions please refer to our Author Guidelines .

- When you resubmit your manuscript, please download our CHECKLIST (<https://bit.ly/EMBOPressAuthorChecklist>) and include the completed form in your submission. *Please note* that the Author Checklist will be published alongside the paper as part of the transparent process (<https://www.embopress.org/page/journal/17444292/authorguide#transparentprocess>)

Please resubmit your revised manuscript online, with a covering letter listing amendments and responses to each point raised by the referees. Please resubmit the paper ****within one month**** and ideally as soon as possible. If we do not receive the revised manuscript within this time period, the file might be closed and any subsequent resubmission would be treated as a new manuscript. Please use the Manuscript Number (above) in all correspondence.

Click on the link below to submit your revised paper.

Kind regards,

Maria

Maria Polychronidou, PhD
Senior Editor
Molecular Systems Biology

If you do choose to resubmit, please click on the link below to submit the revision online before 6th Mar 2024.

IMPORTANT:

See also figure legend guidelines: <https://www.embopress.org/page/journal/17444292/authorguide#figureformat>

- Please note that corresponding authors are required to supply an ORCID ID for their name upon submission of a revised manuscript (EMBO Press signed a joint statement to encourage ORCID adoption).

(<https://www.embopress.org/page/journal/17444292/authorguide#editorialprocess>)

Currently, our records indicate that the ORCID for your account is 0000-0003-3028-6866.

Link Not Available

***** PLEASE NOTE ***** As part of the EMBO Press transparent editorial process initiative (see our Editorial at <https://dx.doi.org/10.1038/msb.2010.72> , Molecular Systems Biology will publish online a Review Process File to accompany accepted manuscripts. When preparing your letter of response, please be aware that in the event of acceptance, your cover letter/point-by-point document will be included as part of this File, which will be available to the scientific community. More information about this initiative is available in our Instructions to Authors. If you have any questions about this initiative, please contact the editorial office (msb@embo.org).

Reviewer #1:

This paper describes a mathematical model for the evolution of homo-and hetero-dimeric interactions after gene duplication.

Because dimers are very common in our structural and interactomics databases, this is a very important question. In particular, this paper tackles the question whether particular patterns in the empirical data can appropriately explained through neutral processes favouring particular outcomes. To do this, the authors use their model to investigate how changes in the parameters of this model can shift the likelihood of different outcomes (whether homo- or heterodimers) dominate after the duplication.

I find this paper very thorough and illuminating. It contains a very thorough and insightful treatment of prior work in the field. The mathematical formalism developed in this paper is useful beyond the confines of this particular study. The authors are also extremely careful to state the limitations of their work. As is the case with all simulation studies, there remains some ambiguity about the realism of some of the underlying assumptions about distributions of mutational effects in particular, but these are clearly acknowledged. Overall, I am very enthusiastic about this study. I hope that in future it might inspire experimental tests of this model on a sufficiently large scale to verify the conclusions derived here.

I have a number of small queries for the authors:

In general, the authors describe the number of mutations necessary for some particular outcome as an absolute number. I think it would be helpful to also express them as a fraction of the length of the protein. Or perhaps even as something similar to a branch length, such that the reader can assess if double substitutions are necessary for some of these outcomes.

One aspect the paper does not discuss (or perhaps I just missed it) is the possibility of dominant negative effects in the evolution such paralogs. Briddgham et al 2008 discuss this possibility for example. Might the authors perhaps speculate in their discussion how this could influence things?

In Figure S3, could the authors please also show histograms of the two individual variables along the top and side of the 2D graphs? This would make it a bit easier to assess the mutational distributions of the individual parameters.

Line 236-237 - text makes it sound like this is a kinetic phenomenon. It is not and the supplemental figure correctly describes this as an equilibrium effect.

Line 305-313 - Great idea to introduce correlations between parameters, but could the authors please provide some sort of justification for these values?

The authors are very thorough and exhaustive in their literature references, but they may consider citing Elizabeth Kaltenecker's very relevant work somewhere.

Reviewer #2:

Cisneros and colleagues use a biophysical modeling approach to understand how gene duplication can result in previously homodimeric complexes being replaced by heterodimeric complexes formed by paralog pairs, even in the absence of any fitness gain resulting from the heterodimeric complex. Their results suggest that, under neutral evolution, heterodimers will generally be favored. This suggests that protein-protein interaction networks may increase in complexity post-duplication even in the absence of any new or improved functions.

The observation that gene duplication can result in increased complexity without fitness gain is consistent with previous work from the authors and others. The primary novelty of the study is in the biophysical modeling approach used and in the systematic evaluation of factors that influence the balance between heteromeric and homomeric assemblies. In particular the authors demonstrate that the relative concentrations of different dimers are highly dependent on binding affinities. In further analyses the authors show that differences in the synthesis rate and activity of different paralogs can counteract the tendency for heterodimeric complexes, providing an explanation for the retention of some homodimeric complexes.

The study is well done - the conclusions are clear, as are the limitations of the analysis. The results will likely be of interest to those studying the structure and evolution of molecular interaction networks and the consequences of gene duplication.

I have only minor comments.

- The authors note that different structures are associated with different outcomes (Figure 2B) and evaluate a number of structural features to see what influences this (p13 top). No clear pattern emerges. It would be useful to have more discussion of what might cause the significant differences between structures.
- Main text figures look fine but the supplemental images are very pixelated (PDF conversion issue?)

Reviewer #3:

This paper is a nice exploration of what happens to homodimers after gene duplication. The base case of neutral evolution suggests that these homodimers should frequently turn into heterodimers, yet evidence from natural organisms does not support this scenario. Therefore, there must be selection against heterodimers.

Overall, the paper is well written and clear, and I don't have much in terms of comments or suggestions. I would only suggest you cite Teufel et al 2019 (<https://academic.oup.com/mbe/article/36/2/304/5182502>) as relevant prior work. In particular, I think some of the concepts explored in that paper, such as duplicated genes driving the evolution of unduplicated binding partners, may be relevant for future explorations of your research question. However, I want to make clear that I don't see a need to add additional scenarios in your current paper.

The other suggestion I would make is to explore whether the paper can be shortened a bit, or the figures simplified. I acknowledge that you have already put a lot of material into the supplement, but I still felt the paper was wordy at times. I don't feel strongly about this though, consider it as an optional suggestion that you're welcome to disagree with.

UNIVERSITÉ
LAVAL

INSTITUT DE
BIOLOGIE INTÉGRATIVE
ET DES SYSTÈMES
IBIS

PROTEO
FONCTION | INGÉNIERIE | APPLICATIONS
DES PROTÉINES

Please find below our detailed answers to each of the points raised by the reviewers. To facilitate the review process, we have marked all changes to the manuscript in blue and highlighted text that responds to the reviewers' comments in yellow. We also include the new versions of relevant figures to reflect changes proposed by the reviewers.

Similarly, we have now organized the supplementary materials as requested. We selected several of the supplementary figures to be included in the Expanded View format and included the rest of them in an appropriately formatted Appendix file. Finally, we include the standfirst text and the synopsis image below.

Dear Christian,

Thank you again for submitting your work to Molecular Systems Biology. We have now heard back from the three reviewers who agreed to evaluate your study. As you will see below, the reviewers find the study interesting and are quite supportive. They do however raise a series of (mostly minor) concerns, which we would ask you to address in a revision.

I think that the reviewers' recommendations are clear and seem straightforward to address. I therefore see no need to repeat any of the comments listed below. All issues raised by the referees would need to be satisfactorily addressed. Please let me know in case you would like to discuss in further detail any of the issues raised, I would be happy to schedule a call.

On a more editorial level, we would ask you to address the following points:

- Please provide a .doc version of the manuscript text (including legends for the main figures) and individual production quality figure files for the main Figures (one file per figure).
- We have replaced Supplementary Information by the Expanded View (EV format). In this case, all additional figures and Tables can be included in a PDF called Appendix. Appendix figures and Tables should be labelled and called out as: "Appendix Figure S1, Appendix Figure S2... Appendix Table S1..." etc. Each legend should be below the corresponding Figure/Table in the Appendix. Please include a Table of Contents in the

UNIVERSITÉ
LAVAL

INSTITUT DE
BIOLOGIE INTÉGRATIVE
ET DES SYSTÈMES
IBIS

PROTEO
FONCTION | INGÉNIERIE | APPLICATIONS
DES PROTÉINES

crdm.ul
CENTRE DE RECHERCHE
EN DONNÉES MASSIVES
DE L'UNIVERSITÉ LAVAL

beginning of the Appendix. For detailed instructions regarding expanded view please refer to our Author Guidelines:

<https://www.embopress.org/page/journal/17444292/authorguide#expandedview>.

- Supplementary Files 1 and 2 should be included in the Appendix (e.g. as Appendix Notes 1 and 2).
- Table S1 is rather complex, it should be provided as Dataset EV1. Please include a description of the Dataset in a separate sheet in the xls file.
- Table S2 should be provided as Table EV1.
- Please provide a "standfirst text" summarizing the study in one or two sentences (approximately 250 characters), three to four "bullet points" highlighting the main findings and a "synopsis image" (550px width and max 400px height, jpeg format) to highlight the paper on our homepage.

Duplicated self-interacting proteins can interact with themselves (homomers) or one another (heteromers). To understand whether natural selection is required to keep homomers over heteromers (or vice versa), we simulate the evolution of such duplicated proteins in the absence of new functions.

- The dynamic equilibrium of homo- and heteromers is given by physical parameters, such as protein folding energy and binding affinities.
- Simulations of the evolution of homodimers from actual structures show a trend toward the increase of the relative concentration of the heterodimer, even when there is no inherent advantage of such an increase.
- The magnitude of the increase in the concentration of heterodimers is associated with mutational biases, that is, an asymmetry with respect to the effects of mutations on homo- and heterodimer binding affinities.
- The bias toward heterodimers can be counterbalanced by changes in the protein synthesis rates or the specific activities of the dimers.

Ancestral self-interacting protein

Gene duplication

Dynamic equilibrium of self- and hetero-interactions

Mutation accumulation

Heteromers dominate	Balanced scenario	Homomers dominate
A A <15%	A A 20-30%	A A >40%
A B >70%	A B 40-60%	A B <20%
B B <15%	B B 20-30%	B B >40%

Relative probability of each outcome in the absence of new functions

- All Materials and Methods need to be described in the main text. We would encourage you to use 'Structured Methods', our new Materials and Methods format. According to this format, the Material and Methods section should include a Reagents and Tools Table (listing key reagents, experimental models, software and relevant equipment and including their sources and relevant identifiers) followed by a Methods and Protocols section in which we encourage the authors to describe their methods using a step-by-step protocol format with bullet points, to facilitate the adoption of the methodologies across labs. More information on how to adhere to this format as well as downloadable templates (.doc or .xls) for the Reagents and Tools Table can be found in our author guidelines:

UNIVERSITÉ
LAVAL

INSTITUT DE
BIOLOGIE INTÉGRATIVE
ET DES SYSTÈMES
IBIS

PROTEO
FONCTION | INGÉNIEURIE APPLICATIONS
DES PROTÉINES

<https://www.embopress.org/page/journal/17444292/authorguide#textformat>. An example of a Method paper with Structured Methods can be found here: <https://www.embopress.org/doi/10.15252/msb.20178071>.

- Please include a "Disclosure & Competing Interests Statement".

- Please include a "Data availability" section describing how the data, code etc. have been made available. This section needs to be formatted according to the example below:

The datasets and computer code produced in this study are available in the following databases:

- Chip-Seq data: Gene Expression Omnibus GSE46748

(<https://www.ncbi.nlm.nih.gov/geo/query/acc.cgi?acc=GSE46748>)

- Modeling computer scripts: GitHub

(<https://github.com/SysBioChalmers/GECKO/releases/tag/v1.0>)

- [data type]: [full name of the resource] [accession number/identifier] ([doi or URL or identifiers.org/DATABASE:ACCESSION])

- For data quantification: please specify the name of the statistical test used to generate error bars and P values, the number (n) of independent experiments (specify technical or biological replicates) underlying each data point and the test used to calculate p-values in each figure legend. The figure legends should contain a basic description of n, P and the test applied. Graphs must include a description of the bars and the error bars (s.d., s.e.m.).

- Molecular Systems Biology supports formal data citations in the Reference list, to cite previously published datasets. In addition to citing the original papers that reported the data, we encourage you to also cite the relevant datasets directly in the Reference list. In the text, references to datasets are included as "Data ref: Smith et al, 2001" or "Data ref: NCBI Sequence Read Archive PRJNA342805, 2017". In the Reference list, data citations are very similar to normal literature references but must be labeled with "[DATASET]" at the end of the reference. For detailed instructions please refer to our Author Guidelines <https://www.embopress.org/page/journal/17444292/authorguide#referencesformat.k>

- When you resubmit your manuscript, please download our CHECKLIST

(<https://bit.ly/EMBOPressAuthorChecklist>) and include the completed form in your submission. *Please note* that the Author Checklist will be published alongside the paper as part of the transparent process

(<https://www.embopress.org/page/journal/17444292/authorguide#transparentprocess>)

UNIVERSITÉ
LAVAL

INSTITUT DE
BIOLOGIE INTÉGRATIVE
ET DES SYSTÈMES
IBIS

PROTEO
FONCTION | INGÉNIERIE APPLICATIONS
DES PROTÉINES

crdm.ul
CENTRE DE RECHERCHE
EN DONNÉES MASSIVES
DE L'UNIVERSITÉ LAVAL

Please resubmit your revised manuscript online, with a covering letter listing amendments and responses to each point raised by the referees. Please resubmit the paper ****within one month**** and ideally as soon as possible. If we do not receive the revised manuscript within this time period, the file might be closed and any subsequent resubmission would be treated as a new manuscript. Please use the Manuscript Number (above) in all correspondence.

Click on the link below to submit your revised paper.

Kind regards,

Maria

Maria Polychronidou, PhD
Senior Editor
Molecular Systems Biology

If you do choose to resubmit, please click on the link below to submit the revision online before 6th Mar 2024.

IMPORTANT:

See also figure legend guidelines:

<https://www.embopress.org/page/journal/17444292/authorguide#figureformat>

UNIVERSITÉ
LAVAL

INSTITUT DE
BIOLOGIE INTÉGRATIVE
ET DES SYSTÈMES
IBIS

PROTEO
FONCTION | INGÉNIERIE | APPLICATIONS
DES PROTÉINES

crdm.ul
CENTRE DE RECHERCHE
EN DONNÉES MASSIVES
DE L'UNIVERSITÉ LAVAL

- Please note that corresponding authors are required to supply an ORCID ID for their name upon submission of a revised manuscript (EMBO Press signed a joint statement to encourage ORCID adoption).

(<https://www.embopress.org/page/journal/17444292/authorguide#editorialprocess>)

Currently, our records indicate that the ORCID for your account is 0000-0003-3028-6866.

<https://msb.msubmit.net/cgi-bin/main.plex?el=A7Ik7BAzQ3A5BdKU2Bh6B9ftdIMhaZr2RQJI4kO03kTnoKQY>

*** PLEASE NOTE *** As part of the EMBO Press transparent editorial process initiative (see our Editorial at <https://dx.doi.org/10.1038/msb.2010.72> , Molecular Systems Biology will publish online a Review Process File to accompany accepted manuscripts. When preparing your letter of response, please be aware that in the event of acceptance, your cover letter/point-by-point document will be included as part of this File, which will be available to the scientific community. More information about this initiative is available in our Instructions to Authors. If you have any questions about this initiative, please contact the editorial office (msb@embo.org).

Reviewer #1:

This paper describes a mathematical model for the evolution of homo- and hetero-dimeric interactions after gene duplication. Because dimers are very common in our structural and interactomics databases, this is a very important question. In particular, this paper tackles the question whether particular patterns in the empirical data can appropriately be explained through neutral processes favouring particular outcomes. To do this, the authors use their model to investigate how changes in the parameters of this model can shift the likelihood of different outcomes (whether homo- or heterodimers) dominate after the duplication.

I find this paper very thorough and illuminating. It contains a very thorough and insightful treatment of prior work in the field. The mathematical formalism developed in this paper is useful beyond the confines of this particular study. The authors are also extremely careful to state the limitations of their work. As is the case with all simulation studies, there remains some ambiguity about the realism of some of the underlying assumptions

UNIVERSITÉ
LAVAL

INSTITUT DE
BIOLOGIE INTÉGRATIVE
ET DES SYSTÈMES
IBIS

PROTEO
FONCTION | INGÉNIERIE APPLICATIONS
DES PROTÉINES

about distributions of mutational effects in particular, but these are clearly acknowledged. Overall, I am very enthusiastic about this study. I hope that in future it might inspire experimental tests of this model on a sufficiently large scale to verify the conclusions derived here.

We thank the reviewer for their positive comments about our work. Likewise, we are enthusiastic about the possibility of experimental tests of this model.

I have a number of small queries for the authors:

In general, the authors describe the number of mutations necessary for some particular outcome as an absolute number. I think it would be helpful to also express them as a fraction of the length of the protein. Or perhaps even as something similar to a branch length, such that the reader can assess if double substitutions are necessary for some of these outcomes.

Yes, this is an important point considering that 200 mutations could drastically alter the sequence of very small proteins. We now have some discussion about the range of mutation count / protein length ratios for the proteins in our dataset. Furthermore, we would like to point out that we compared the sequences of the two proteins at the end of the simulations against each other and against the starting sequences. Even for the smallest proteins in our dataset, sequence identity between paralogs stayed at around 30% and sequence identity with the starting sequences remained around 50%. This result suggests that our distributions of mutational effects capture structural constraints on sequence, as mutations in some sites never fix in the simulations.

The corresponding text is in lines 331-339 of the manuscript and reads as follows: “Some of the tested proteins are shorter than 200 residues (mean = 240, interquartile range = [157.8, 332], minimum length of 66). Although this implies that 200 mutations could be enough to mutate every position in the sequence, the sequences of paralogs maintained about 50% identity with the WT sequence and more than 30% sequence identity with one another by the end of the simulations. These measures are consistent with real-world scenarios where more than half of *Saccharomyces cerevisiae* and *Escherichia coli* paralogous pairs exhibit $\geq 30\%$ identity (Mallik & Tawfik, 2020) and suggest our model captures structural constraints in the relative mutability of different positions.”

One aspect the paper does not discuss (or perhaps I just missed it) is the possibility of dominant negative effects in the evolution such paralogs. Bridgman et al 2008 discuss

UNIVERSITÉ
LAVAL

INSTITUT DE
BIOLOGIE INTÉGRATIVE
ET DES SYSTÈMES
IBIS

PROTEO
FONCTION | INGÉNIERIE APPLICATIONS
DES PROTÉINES

crdm.ul
CENTRE DE RECHERCHE
EN DONNÉES MASSIVES
DE L'UNIVERSITÉ LAVAL

this possibility for example. Might the authors perhaps speculate in their discussion how this could influence things?

We thank the reviewer for raising this point. A mutation with a dominant negative effect would compromise the catalytic activity of the homodimer of the mutated paralog and the heterodimer. Thus, a dominant negative effect would result in a significant loss of the total activity in the system, depending on the equilibrium constants. Also, we could speculate that dominant negative mutations may not necessarily imply total loss of activity and result in different decreases of specific activity for the homodimer and the heterodimer. We initially excluded this point from our simulations because it would require setting additional parameters: a probability factor for the mutation having a double negative effect and the magnitude of the decrease in the specific activity of the two mentioned dimers. While we can explore the effect of different values for these parameters on the simulations, we do not have reference typical values. Moreover, we would expect these parameters to be different for every protein in the dataset. We therefore did not dig further into this direction for the current manuscript but we would like to examine that in the future.

All this considered, we added the following text in the discussion (lines 800-804) to provide some perspectives about dominant negative effects:

“A special case of this evolution could be the emergence of dominant negative mutations, which would inactivate the homodimer of the mutated paralog and the heterodimer (Veitia 2007). Depending on the optimal activity, this reduction in activity could either lead to selection against the heterodimer to restore activity or lead to the emergence of new regulatory roles for the heterodimer (Bridgham et al. 2008).

In Figure S3, could the authors please also show histograms of the two individual variables along the top and side of the 2D graphs? This would make it a bit easier to assess the mutational distributions of the individual parameters.”

We have now added the requested histograms to improve data visualization in Figure S3. Please keep in mind that now that we have formatted our files for Molecular Systems Biology, Figure S3 became Figure EV1. Please find the new version of this figure below:

Line 236-237 - text makes it sound like this is a kinetic phenomenon. It is not and the supplemental figure correctly describes this as an equilibrium effect.

We thank the reviewer for pointing this out. We have now rewritten the corresponding section (lines 234-238) as follows:

“Interestingly, there are regions in the solution space in which most of the available subunits assembled into homodimers although one of the homodimers had a slightly weaker binding affinity than the heterodimer (Appendix Figure S1E). In this case, equilibrium favors the strongest homodimer. As a result, heterodimers are depleted in the system and the second protein forms its respective homodimer.”

Line 305-313 - Great idea to introduce correlations between parameters, but could the authors please provide some sort of justification for these values?

The magnitude of the correlations between mutational effects on binding affinity and folding energy was derived from the actual FoldX data. We have now modified the corresponding sentences (lines 304-312) to make it clearer:

“Since mutations can affect both binding affinity and folding free energy, we derive multivariate normal distributions for the effects of mutations based on the pooled data. These distributions have the following parameters, all in kcal/mol: for $\Delta\Delta G_{\text{bind, HET}}$, mean = 0.2, standard deviation = 1.2 (denoted as $N(0.2, 1.2)$); for $\Delta\Delta G_{\text{bind, HM}}$, $N(0.4, 2.4)$; and for $\Delta\Delta G_{\text{fold}}$, $N(2.6, 4.6)$. These FoldX derived data also revealed the extent of correlations between mutational effects on folding energy and binding affinity: $r = 0.9$ between $\Delta\Delta G_{\text{bind, HET}}$ and $\Delta\Delta G_{\text{bind, HM}}$ (Figure EV1A), r

UNIVERSITÉ
LAVAL

INSTITUT DE
BIOLOGIE INTÉGRATIVE
ET DES SYSTÈMES
IBIS

PROTEO
FONCTION | INGÉNIERIE APPLICATIONS
DES PROTÉINES

=
0.3

between $\Delta\Delta G_{\text{bind, HET}}$ and $\Delta\Delta G_{\text{fold}}$ (Figure EV1B), and $r = 0.3$ between $\Delta\Delta G_{\text{bind, HM}}$ and $\Delta\Delta G_{\text{fold}}$ (Figure EV1C). As such, we imposed these correlations in our parametric simulations.”

The authors are very thorough and exhaustive in their literature references, but they may consider citing Elizabeth Kaltenecker's very relevant work somewhere.

We thank the reviewer for this suggestion. We have now added references to her most relevant works in the introduction and discussion sections of the manuscript (lines 57, 67, 137, 741-743).

Reviewer #2:

Cisneros and colleagues use a biophysical modeling approach to understand how gene duplication can result in previously homodimeric complexes being replaced by heterodimeric complexes formed by paralog pairs, even in the absence of any fitness gain resulting from the heterodimeric complex. Their results suggest that, under neutral evolution, heterodimers will generally be favored. This suggests that protein-protein interaction networks may increase in complexity post-duplication even in the absence of any new or improved functions.

The observation that gene duplication can result in increased complexity without fitness gain is consistent with previous work from the authors and others. The primary novelty of the study is in the biophysical modeling approach used and in the systematic evaluation of factors that influence the balance between heteromeric and homomeric assemblies. In particular the authors demonstrate that the relative concentrations of different dimers are highly dependent on binding affinities. In further analyses the authors show that differences in the synthesis rate and activity of different paralogs can counteract the tendency for heterodimeric complexes, providing an explanation for the retention of some homodimeric complexes.

The study is well done - the conclusions are clear, as are the limitations of the analysis. The results will likely be of interest to those studying the structure and evolution of molecular interaction networks and the consequences of gene duplication.

We thank the reviewer for their positive evaluation of our work.

I have only minor comments.

UNIVERSITÉ
LAVAL

INSTITUT DE
BIOLOGIE INTÉGRATIVE
ET DES SYSTÈMES
IBIS

PROTEO
FONCTION | INGÉNIERIE | APPLICATIONS
DES PROTÉINES

- The authors note that different structures are associated with different outcomes (Figure 2B) and evaluate a number of structural features to see what influences this (p13 top). No clear pattern emerges. It would be useful to have more discussion of what might cause the significant differences between structures.

This is an important point. One of the factors that could contribute to the difficulty in identifying such factors is the diversity in our dataset. We aimed to make it broad to include different types of protein architectures, we are left with small sample sizes for each of them. Moreover, when we compare different proteins, multiple structural features change at the same time (i. e. protein architecture, secondary structure composition, interface size, symmetry). As such, it becomes hard to associate any difference in the observed distribution of mutational effects with any of these structural features. We believe that future studies should try to distinguish the contributions of these features using a controlled set of PDB structures that allow isolating the effect of individual features.

We have now added the following sentences (lines 771-775) in the discussion:

“Future work should focus on experimental characterizations of the distribution of mutational effects of protein structures. While our analyses point to the existence and effects of mutational biases, we could not fully establish their structural determinants. Comparing larger-scale datasets might allow isolating the effects of individual structural features on the distribution of mutational effects.”

- Main text figures look fine but the supplemental images are very pixelated (PDF conversion issue?)

We thank the reviewer for pointing this out. We have now revised the quality of our supplementary figures in the document.

Reviewer #3:

This paper is a nice exploration of what happens to homodimers after gene duplication. The base case of neutral evolution suggests that these homodimers should frequently turn into heterodimers, yet evidence from natural organisms does not support this scenario. Therefore, there must be selection against heterodimers.

We thank the reviewer for their positive comments.

UNIVERSITÉ
LAVAL

INSTITUT DE
BIOLOGIE INTÉGRATIVE
ET DES SYSTÈMES
IBIS

PROTEO
FONCTION | INGÉNIERIE | APPLICATIONS
DES PROTÉINES

crdm.ul
CENTRE DE RECHERCHE
EN DONNÉES MASSIVES
DE L'UNIVERSITÉ LAVAL

Overall, the paper is well written and clear, and I don't have much in terms of comments or suggestions. I would only suggest you cite Teufel et al 2019 (<https://academic.oup.com/mbe/article/36/2/304/5182502>) as relevant prior work. In particular, I think some of the concepts explored in that paper, such as duplicated genes driving the evolution of unduplicated binding partners, may be relevant for future explorations of your research question. However, I want to make clear that I don't see a need to add additional scenarios in your current paper.

We thank the reviewer for pointing out this relevant paper. We have now included a brief discussion of these ideas (lines 693-699) in the main text:

“Furthermore, mutational effects on binding for homodimers and heterodimers are highly correlated. As a result, an avenue for removing one of the dimers could be the initial transient destabilization of both followed by the subsequent restabilization of the favored interaction, as observed for duplicated proteins forming heteromers with a common partner (Teufel et al. 2019). However, our FoldX predictions suggest that there is a small percentage of mutations that could directly stabilize one complex while destabilizing the other (Figure 3A, Figure EV1A).”

The other suggestion I would make is to explore whether the paper can be shortened a bit, or the figures simplified. I acknowledge that you have already put a lot of material into the supplement, but I still felt the paper was wordy at times. I don't feel strongly about this though, consider it as an optional suggestion that you're welcome to disagree with.

We considered this comment from the reviewer. However, we preferred to not shorten the main text to avoid compromising the clarity of our explanation of the model and how the different parameters interact with one another. Accordingly, since the paper is indeed rather long, we aimed to be concise with our responses to the points raised by other reviewers.

23rd Feb 2024

Manuscript Number: MSB-2024-12212R

Title: Mutational biases favor complexity increases in protein interaction networks after gene duplication

Dear Christian,

Thank you for sending us your revised manuscript. We have evaluated your revision and we think that all issues raised by the reviewers have been addressed. I am glad to inform you that we can soon accept the manuscript for publication, pending some final editorial requests listed below.

- Our data editors have noted that the following needs to be edited in the figure legends:
 - Please indicate the statistical test used for data analysis in the legends of figures 3e-f; EV 2b; EV 3d-e.
 - The box plots need to be defined in terms of minima, maxima, center, bounds of box and whiskers, and percentile in the legends of figures EV 2e-f; EV 5c.
- The legends for the EV Figures need to be included in the main text, after the legends for the main figures.
- Please remove the numbering from the Methods (sub-)sections.
- In the figure legends of the Appendix PDF, the figures should be labeled as "Appendix Figure 1, Appendix Figure 2 etc." (instead of "Figure S1, Figure S2 etc.")
- The synopsis image does not display well at the final size required. Please resupply the image as a jpg or png at the required final size (it needs to be exactly 550 px wide, and the height ideally < 500 px), ensuring that all labels are legible. Reorganizing the image in a more landscape orientation might work better.
- The funding information provided in the manuscript text need to match the information entered in the online submission system. Currently the following is missing from the submission system: Merit Scholarship Program for Foreign Studies (PBEEE); MEES. The grant numbers 290237 and IT28316 are missing from the manuscript text.
- Please remove the 'Authors Contributions' from the manuscript. The 'Author Contributions' section is replaced by the CRediT contributor roles taxonomy to specify the contributions of each author in the journal submission system. Please use the free text box in the 'author information' section of the online submission system to provide more detailed descriptions if needed (e.g., 'X provided intracellular Ca⁺⁺ measurements in fig Y').

Please resubmit your revised manuscript online ****within one month**** and ideally as soon as possible. If we do not receive the revised manuscript within this time period, the file might be closed and any subsequent resubmission would be treated as a new manuscript. Please use the Manuscript Number (above) in all correspondence.

Click on the link below to submit your revised paper.

Kind regards,

Maria

Maria Polychronidou, PhD
Senior Editor
Molecular Systems Biology

If you do choose to resubmit, please click on the link below to submit the revision online before 24th Mar 2024.

IMPORTANT:

Please note that corresponding authors are required to supply an ORCID ID for their name upon submission of a revised manuscript (EMBO Press signed a joint statement to encourage ORCID adoption).

(<https://www.embopress.org/page/journal/17444292/authorguide#editorialprocess>)

Currently, our records indicate that the ORCID for your account is 0000-0003-3028-6866.

Link Not Available

*** PLEASE NOTE *** As part of the EMBO Press transparent editorial process initiative (see our Editorial at <https://dx.doi.org/10.1038/msb.2010.72> , Molecular Systems Biology will publish online a Review Process File to accompany accepted manuscripts. When preparing your letter of response, please be aware that in the event of acceptance, your cover letter/point-by-point document will be included as part of this File, which will be available to the scientific community. More information about this initiative is available in our Instructions to Authors. If you have any questions about this initiative, please contact the editorial office (msb@embo.org).

All editorial and formatting issues were resolved by the authors.

28th Feb 2024

Manuscript number: MSB-2024-12212RR

Title: Mutational biases favor complexity increases in protein interaction networks after gene duplication

Dear Christian,

Thank you again for sending us your revised manuscript. We are now satisfied with the modifications made and I am pleased to inform you that your paper has been accepted for publication.

Kind regards,

Maria

Maria Polychronidou, PhD
Senior Editor
Molecular Systems Biology
